



# New approach to evaluate satellite derived XCO₂ over oceans by integrating ship and aircraft observations

Astrid Müller[1], Hiroshi Tanimoto[1], Takafumi Sugita[1], Toshinobu Machida[1], Shin-ichiro Nakaoka[1], Prabir K. Patra[2], Joshua Laughner[3], David Crisp[4]

1 National Institute for Environmental Studies, Tsukuba, Japan
2 Japan Agency for Marine-Earth Science and Technology, Yokohama, Japan
3 California Institute of Technology, Pasadena, CA, USA
Jet Propulsion Laboratory/California Institute of Technology, Pasadena, CA, USA

*Correspondence and request for material should be addressed to*:
Hiroshi Tanimoto (tanimoto@nies.go.jp), Astrid Müller (mueller.astrid@nies.go.jp)

**Abstract.** Satellite observations provide spatially-resolved global estimates of column-averaged mixing ratios of $CO_2$ ($XCO_2$)

over the Earth's surface. The accuracy of these datasets can be validated against reliable standards in some areas, but other areas remain inaccessible. To date, limited reference data over oceans hinders successful uncertainty quantification or bias correction efforts, and precludes reliable conclusions about changes in the carbon cycle in some regions. Here, we propose a new approach to analyze and evaluate seasonal, interannual and latitudinal variations of $XCO_2$ over oceans by integrating cargo-ship (SOOP, Ship Of Opportunity) and commercial aircraft (CONTRAIL, Comprehensive Observation Network for

Trace gases by Airliner) observations with the aid of state-of-the art atmospheric chemistry-transport model calculations. The consistency of the "in situ based column-averaged $CO_2$" dataset (in situ $XCO_2$) with satellite estimates was analyzed over the Western Pacific between 2014 and 2017, and its utility as reference dataset evaluated. Our results demonstrate that the new dataset accurately captures seasonal and interannual variations of $CO_2$. Retrievals of $XCO_2$ over the ocean from GOSAT (Greenhouse gases observing satellite: NIES v02.75, National Institute for Environmental Studies; ACOS v7.3, Atmospheric

$CO_2$ Observation from Space) and OCO-2 (Orbiting Carbon Observatory, v9r) observations show a negative bias of about 1 parts per million (ppm) in northern midlatitudes, which was attributed to measurement uncertainties of the satellite observations. The NIES retrieval had higher consistency with in situ $XCO_2$ at midlatitudes as compared to the other retrievals. At low latitudes, it shows many fewer valid data and high scatter, such that ACOS and OCO-2 appear to provide a better representation of the carbon cycle. At different times, the seasonal cycles of all three retrievals show positive phase shifts of

one month relative to the in situ data. The study indicates that even if the retrievals complement each other, remaining uncertainties limit the accurate interpretation of spatiotemporal changes in $CO_2$ fluxes. A continuous long-term $XCO_2$ dataset with wide latitudinal coverage based on the new approach has a great potential as a robust reference dataset for $XCO_2$ and can help to better understand changes in the carbon cycle in response to climate change using satellite observations.



## 1 Introduction

Efforts to control the accelerated increase of carbon dioxide ($CO_2$) in the atmosphere became a serious international task in the last decades. $CO_2$ is the most important anthropogenic greenhouse gas (GHG). Since the beginning of the Industrial Era in the 1750s, fossil fuel combustion and other human activities have increased the atmospheric concentration of $CO_2$ from approximately 277 ppm to more than 407 ppm in 2018 (Friedlingstein et al., 2019). On average, less than half of the anthropogenic $CO_2$ emitted each year stays in the atmosphere, as the ocean and land each capture approximately one-fourth

(Friedlingstein et al., 2019). Seasonal changes in $CO_2$ uptake and release alter the fraction of atmospheric $CO_2$ substantially and lead to year-to-year variations, which are not yet fully understood (e.g. Friedlingstein et al., 2019; Intergovernmental Panel on Climate Change (IPCC), 2013). As the carbon cycle responds to a changing climate, a comprehensive understanding of changes in $CO_2$ sources and sinks is crucial to the implementation of effective strategies for reducing global warming.

In situ measurements from ground-based networks and aircraft campaigns provide precise information on local $CO_2$

concentrations. There are now more than 100 surface measurement sites around the globe, but most are located on land in north America and Europe, and some in the East Asia and Oceania (e.g., Crowell et al., 2019; Hakkarainen et al., 2019). Very few sites are located over the open oceans, even though 70% of the Earth's surface is covered by water and the ocean is a key element of the global carbon cycle. The uneven distribution and limited spatial coverage of in situ measurements makes it impossible to infer $CO_2$ fluxes between the surface and the atmosphere on regional to global scales (Canadell et al., 2011;

Chevallier et al., 2010, 2011). Space-based remote sensing measurements are complementing in situ observations. Their high spatial and temporal coverage allows observation of changes in atmospheric $CO_2$ mixing ratios even in regions with poor in situ coverage (Baker et al., 2010, Crisp et al., 2012). By collecting high resolution spectra of near infrared (NIR) and shortwave infrared (SWIR) solar radiation reflected from the Earth's surface, satellite observations can yield estimates of the total atmospheric column of $CO_2$. These observations are most sensitive to the lower troposphere where $CO_2$ is most variable (Patra

et al., 2003) and therefore, are able to improves the knowledge on local $CO_2$ emission and sinks (Connor et al., 2008).

Japan's Greenhouse gases Observing Satellite (GOSAT), and the second NASA (National Aeronautics and Space Administration) Orbiting Carbon Observatory (OCO-2) are dedicated to inferring the concentration of GHGs from high-resolution spectra at NIR and SWIR wavelengths. Since their launches in 2009 and 2014, GOSAT and OCO-2 have successfully provided global datasets of column-averaged mixing ratios of $CO_2$ ($XCO_2$). In 2018, GOSAT-2 was launched,

aiming to improve the measurement precision and to overcome anomalies of the spectrometer on board GOSAT (Nakajima et al., 2017). The launch of OCO-3 followed in 2019. Since 2009, NASAs Atmospheric $CO_2$ Observation from Space (ACOS) and GOSAT team work closely together on the analysis of GOSAT observations (Crisp et al., 2012; O'Dell et al., 2012). Comparisons of $XCO_2$ generated by the GOSAT team of the National Institute for Environmental Studies (NIES) (e.g., Yoshida et al., 2013) with that of the ACOS retrieval algorithm are aimed to improve the accuracy of the estimated $XCO_2$.





Variations in the $CO_2$ concentration associated with surface sources and sinks are typically not larger than 1 ppm (0.25%), and annual and seasonal variations of $XCO_2$ are small compared to the mean abundance in the atmosphere (Crisp et al., 2012; Miller et al., 2007). Therefore, a precision of 1–2 ppm for $CO_2$ satellite retrievals is needed (Crisp et al., 2012). Any uncharacterized systematic errors in the retrieval affect the accuracy of $XCO_2$ and limit its utility for carbon cycle studies (Basu et al., 2013). Therefore, extensive validation of satellite $XCO_2$ has been performed, mainly against data of the Total Carbon

Column Observing Network (TCCON) (Wunch et al., 2011), which is a network of ground-based Fourier transform infrared (FTIR) spectrometers. However, TCCON has a very limited number of sites observing open oceans. Between the GOSAT NIES retrieval and TCCON sites near the ocean, a bias of $-1.09 \pm 2.27$ ppm was found (Morino et al., 2020). Negative $XCO_2$ anomalies north and south of the equator are observed in the OCO-2 retrieval over the Pacific Ocean (Hakkarainen et al., 2019). In combination with surface measurements, vertical profiles of $CO_2$ obtained by aircrafts can constrain $XCO_2$ but are

very limited (e.g., Frankenberg et al., 2016; Inoue et al., 2013; Wofsy, 2011; Wofsy et al., 2018). Inoue et al. (2013) found a bias as large as $-1.8$ to $-2.3$ ppm between aircraft-based $XCO_2$ and that from GOSAT NIES at the Pacific Ocean. Comparisons of ACOS GOSAT $XCO_2$ estimates to those from HIAPER Pole-to-Pole Observations (HIPPO) campaigns (Frankenberg et al., 2016) show lower bias ($-0.06$ ppm) and a standard deviation (0.45 ppm). More recent comparisons of OCO-2 $XCO_2$ estimates to in situ measurements from the NASA Atmospheric Tomography Mission reveals a systematic bias of -0.7 ppm over the

tropical Pacific, that is also seen in TCCON data in that region (Kulawik et al., 2019, Atmos. Meas. Tech. Discuss., https://doi.org/10.5194/amt-2019-257). Limited reference data in the tropical and high latitudinal oceans are the reason for major uncertainties in satellite retrievals over these regions. Therefore, variations in $XCO_2$ over ocean sites cannot be reliably captured, but this is necessary for modeling the future climate (e.g., Crowell et al., 2019).

We propose a new approach to analyze and evaluate seasonal, interannual and latitudinal variations of satellite derived $XCO_2$

by integrating cargo-ship and commercial aircraft observations. We use long-term datasets of the dry air mole fraction of $CO_2$ from Japan's CONTRAIL (Comprehensive Observation Network for Trace gases by Airliner) and SOOP (Ship Of Opportunity) project which cover wide latitudinal and longitudinal regions of the Pacific and South China Sea. Together with state-of-the art atmospheric chemistry-transport model calculations (Patra et al., 2018), we calculate in situ based $XCO_2$. The consistency of the spatiotemporal variation of the ship-aircraft based $XCO_2$ with satellite estimates from OCO-2, and two

GOSAT retrievals (NIES, ACOS) is analyzed, and its utility as long-term reference dataset evaluated.

## 2 Observational Data

### 2.1 Aircraft

Japan's Comprehensive Observation Network for Trace gases by Airliner, CONTRAIL, uses commercial aircraft flying between Japan and Europe, Asia, Australia, Hawaii and North America to continuously measure atmospheric $CO_2$ since 2005.

In cooperation with Japan Airlines (JAL), the Continuous $CO_2$ Measuring Equipment (CME) is installed in the forward cargo



compartment on 777-200ER or 777-300ER aircraft (Machida et al., 2008; Umezawa et al., 2018). The CME measures the $CO_2$ dry mole fraction using a non-dispersive infrared gas analyzer (NDIR; LI-840, LI-COR Biogeosciences). Air samples are taken from the air conditioning system of the aircraft. Before the samples are analyzed by the NDIR, a diaphragm pump draws the samples through a drier tube packed with $CO_2$-saturated magnesium perchlorate to remove water vapor. The flow rate and

absolute pressure in the NDIR are kept constant by a mass flow controller and auto pressure controller, respectively.

Two standard gases are introduced into the NDIR every 14 minutes (min) during the ascent and decent portions of the flight and every 62 min during the cruise at 8-12 km height (Machida et al., 2008; Umezawa et al., 2018). Forty seconds (s) after the switch from standard gas to air sample, data are collected as averages of 10 s during the ascent and decent, and 1 min averages during the cruise (~ 15 km horizontal distance) if the standard deviation does not exceed 3 ppm (Umezawa et al., 2018). The

analytical uncertainty of the CME is 0.2 ppm, which was estimated from the comparison with occasional flask sampling, using an automatic air sampling equipment (Matsueda et al., 2008).

In this study, we used CME data v2019.1.0 from flights between Narita and Sydney over the Western Pacific Ocean between 2014 and 2017. Only those data which were obtained below the tropopause height during the cruise at around 11 km altitude are used.

**2.2 Ship**

Commercial cargo Ships of Opportunity (SOOP) have been collecting samples of atmospheric $CO_2$ on cruises since 2001 between Japan and North America, since 2005 between Japan and Australia and New Zealand, and since 2007, between Japan and South East Asia. In this study, we used data collected by the cargo ship Trans Future 5 (Toyofuji Shipping Co., Ltd.), which sails between Japan, Australia, and New Zealand. The dry air mole fraction of $CO_2$ is measured by a NDIR (MOG-701,

Kimoto Electric Co.) every 10 s with an accuracy of 0.1 ppm. The NDIR is installed on top of the bridge at approximately 30 m above sea level (Yamagishi et al., 2012). Samples are drawn into the NDIR through a tube, whose inlet is placed at a location which is not affected by smoke of the ship. Calibration is done every 6 hours by introducing four $CO_2$ standards (360, 380, 400, 420 ppm, Taiyo Nippon Sanso Corporation, Japan).

**2.3 Satellite**

Japan's GOSAT launched in 2009, and NASA's OCO-2 launched in 2014, were developed to characterize the variability of the atmospheric $CO_2$ fraction at regional scales over the globe. Both the OCO-2 grating spectrometer and the Thermal And Near infrared Sensor for carbon Observations – Fourier Transform Spectrometer (TANSO-FTS) instrument on board GOSAT measure the reflected sunlight in three shortwave infrared (SWIR) channels: at around 0.764 μm, which contains significant $O_2$ absorption, at 1.61 μm which contains a weak $CO_2$ absorption band, and at 2.06 μm, containing a strong $CO_2$ absorption

band (Crisp et al., 2017; Kuze et al., 2009). By measuring the amount of light absorbed by $CO_2$ and $O_2$, the column average



CO$_2$ dry air mole fraction (XCO$_2$) is estimated by taking ratio of the total column amounts of CO$_2$ and O$_2$, where O$_2$ provides an estimate for the total column of dry air (Wunch et al., 2011).

When the launch system failed for the first OCO in 2009, the ACOS team modified the retrieval algorithm originally developed for OCO to allow GOSAT retrievals (O'Dell et al., 2012). In this study, we selected level 2 XCO$_2$ data in sun-glint mode from the NIES v02.75 (Yoshida et al., 2013), ACOS v7.3, and OCO-2 v9r retrieval algorithm, all of which were bias corrected. NIES v02.75 uses only cloud-free scenes. For ACOS and OCO-2, we chose data with a good quality flag (quality_flag = 0), which is provided by each algorithm. The ACOS data processing is ongoing and data of version 7.3 are available until June 2016. At the time of writing the manuscript, ACOS version 9 was released. This version is based on a newer version of the GOSAT Level 1 product, which includes extended sun-glint data. An initial comparison between ACOS v7.3 and v9r is included in the supplement (**Fig. A1**) and section 5 Conclusions. In the following, we refer to data obtained by OCO-2 v9r and GOSAT using the retrieval algorithm from NIES v02.75 and ACOS v7.3 simply as "OCO-2", "NIES", and "ACOS", respectively.

## 3 Methodology

### 3.1 Data selection

In order to compare data of all satellite retrievals, we chose the time period from 2014 to 2017, when both GOSAT (NIES, ACOS) and OCO-2 XCO$_2$ products are available. Over the Western Pacific between 40° N and 30° S, we made 10° latitude by 20° longitude wide boxes around the ship and aircraft data in order to obtain enough co-located data for the seasonal and interannual comparison with satellite retrievals **(Fig. 1)**. Within these boxes, no significant latitudinal and longitudinal variation of the CO$_2$ mixing ratio is expected (Sawa et al., 2012). Results of the MIROC-4 (Model for Interdisciplinary Research On Climate Earth System, version 4.0)-based Atmospheric Chemistry Transport model (ACTM) were obtained for each hourly averaged location of the aircraft (details are explained below). All data obtained over land are excluded. The monthly averages of the satellite, in situ, and model datasets are used for the further analysis. In this study, we focus on the results of the latitude ranges 20° N–30° N, 0° N–10° N, and 20° S–10° S, as representative for the northern mid latitude, the equator region, and southern latitudes, respectively.

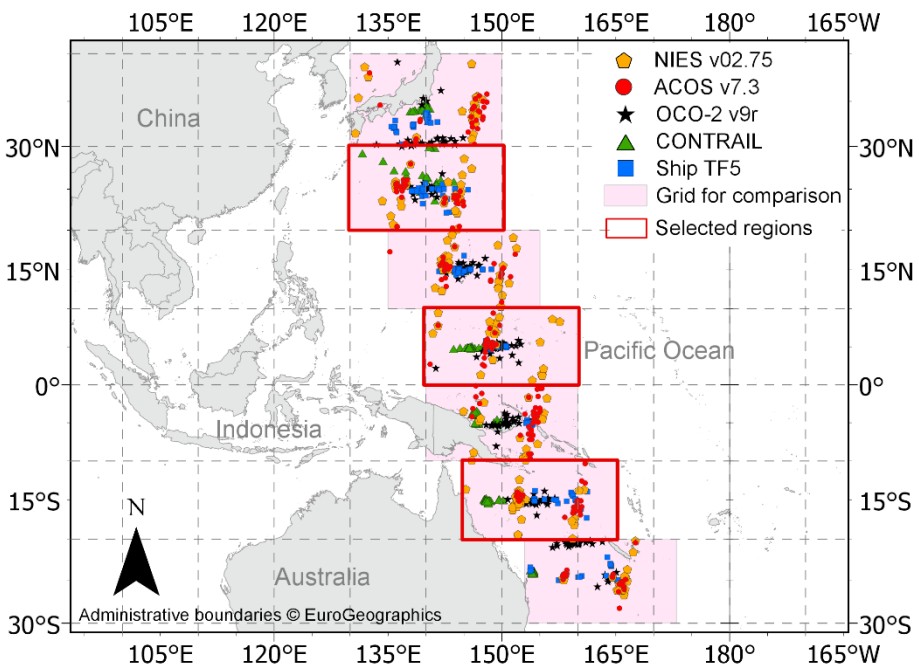

**Figure 1.** Location of monthly averaged data of $CO_2$ from aircraft (CONTRAIL, green triangle), ship (Trans Future 5 - TF5, blue squares), the satellite retrievals from NIES (yellow diamonds), ACOS (red circles), and OCO-2 (black stars) between 2014 and 2017. Selected regions within 10° latitude by 20° longitude boxes are shown in red frames. Administrative boundaries © EuroGeographics.

## 3.2 In situ profile construction and XCO₂ calculation

**Figure 2** shows how atmospheric $CO_2$ profiles are constructed with the aid of ship and aircraft data in order to derive column averaged mixing ratios of $CO_2$. Ship data are extrapolated vertically to ~850 hPa. Previous balloon and aircraft measurements by the HIPPO campaign over the Tropical Eastern and Western Pacific showed $CO_2$ variation of 1 to 2 ppm within the first 2 km above sea-level (Frankenberg et al., 2016; Inai et al., 2018). To account for that variation, we added a ±2 ppm uncertainty to the $CO_2$ estimates of that pressure level. Aircraft data from the cruise portion of the flight, which is usually between 380 and 200 hPa, are selected. These aircraft data are extrapolated down to the lower cruising height limit at 380 hPa, and at 30° N–40° N at 400 hPa. The blended tropopause pressure (TROPPB) is used as upper limit for the extrapolation. It is defined as a combination of a thermal tropopause- and dynamic tropopause pressure (Wilcox et al., 2012). The TROPPB data are extracted from GEOS-FP (Goddard Earth Observing System – forward processing) meteorology data using the python suite "ginput". Ginput was developed to generate a priori vertical mixing ratios of chemical species (e.g., $CO_2$, CO, $CH_4$, $N_2O$) for the open source TCCON retrieval algorithm, GGG2020 (Laughner et al., in prep). Assuming a straight profile between the extrapolated aircraft and ship data, we linearly interpolate in both pressure and volume mixing ratio.

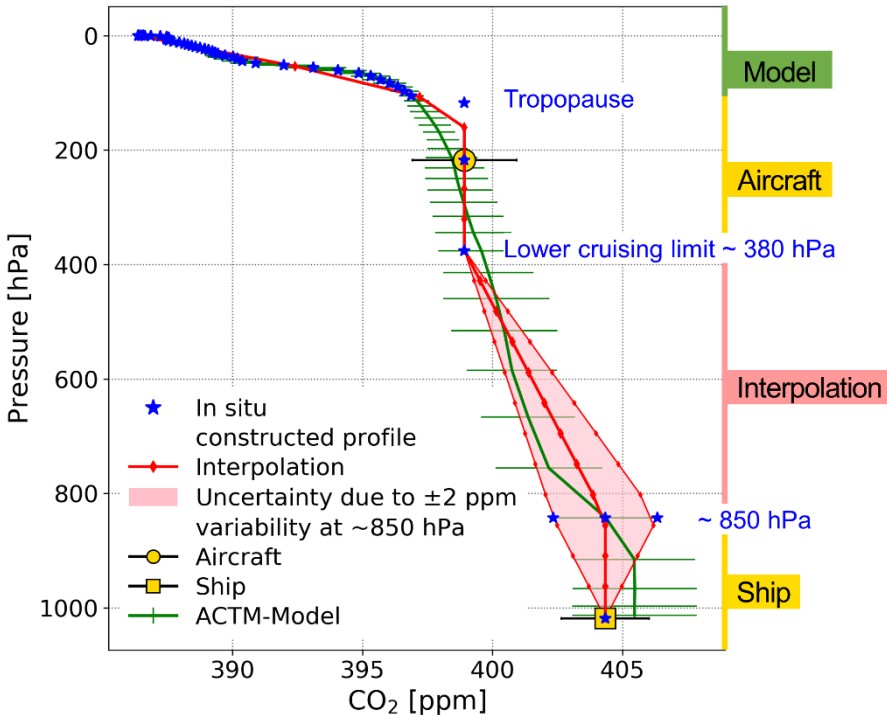

**Figure 2.** Construction of the in situ adjusted $CO_2$ profile (blue) by using ship (SOOP) and aircraft (CONTRAIL) data (yellow) together with the results of the ACTM (green), and the interpolation (red). The example is obtained at the latitude 20° N–30° N, March 2014.

Total column observations in the atmosphere consists of up to 40% air in the stratosphere (Patra et al., 2018). To account for the stratospheric partial column, we used results of the MIROC-4 ACTM (Patra et al., 2018) above the TROPPB **(Fig. 2)**. The details of the MIROC-4 ACTM are described in Patra et al. (2018). In short, the MIROC-4 ACTM uses a hybrid vertical coordinate to resolve gravity wave propagation into the stratosphere. The hybrid coordinate transitions from sigma coordinates at the surface to pressure levels around the tropopause. The ACTMs are nudged with the Japanese 55-year Reanalysis (JRA-55; Kobayashi et al., 2015) for horizontal winds and temperature at Newtonian relaxation times of 1-hour and 5-hours, respectively. Nudging is performed for all the model layers from 2 to 60. In total, 67 vertical layers are used between the Earth's surface and 0.0128 hPa. A high accuracy of the MIROC-4-ACTM is indicated by the agreement of simulated "age of air", which is a diagnostic for atmospheric transport, with that expected from measured sulphur hexafluoride ($SF_6$) and $CO_2$ in the troposphere and stratosphere, respectively (Patra et al., 2018).

To calculate the $XCO_2$ that the satellite would have seen given the $CO_2$ profile constructed from in situ data, we use Eq. (15) of Connor et al. (2008):

$$X_{CO2}^m = X_{CO2}^a + \sum_j h_j \, a_{co2,j} \, (x_m - x_a)_j \qquad (1)$$



Here, $X_{CO2}^m$ is the total column XCO₂ that the satellite would report if it observed the constructed in situ CO₂ profile $x_m$. We refer to $X_{CO2}^m$ as "in situ XCO₂" in the following. $x_m$ is the in situ constructed CO₂ profile (as a true profile). Extracted from the corresponding satellite retrievals, $X_{CO2}^a$ is the a-priori XCO₂ of OCO-2, NIES, and ACOS, respectively, $h_j$ the pressure weighting function, which is the change of atmospheric transmittance with respect to the pressure, $a_{CO2,j}$ is the column averaging kernel, which represents the sensitivity profile to the total column amount, and $x_a$ the a priori CO₂ profile.

Comparison between monthly averages of the calculated in situ XCO₂ using $X_{CO2}^a$, $h_j$, $a_{CO2,j}$, and $x_a$ from the NIES and ACOS files showed agreement within 0.1 ± 0.1 ppm. Because the ACOS retrieval provides a higher number of valid data, we used the parameters from ACOS as representative for the calculation. After May 2016, we use the parameters from NIES due to the temporal limit of the ACOS v7.3 product.

It is noted that in our approach to obtain in situ XCO₂, the usage of model results above the TROPPB introduces little bias for

two reasons. First, the CO₂ mixing ratio at these pressure levels varies much less than that in the middle and lower troposphere since there are no significant CO₂ sources and sinks in the stratosphere. Second, the MIROC4-ACTM is among the best validated stratospheric models (Patra et al., 2018). Furthermore, in a sensitivity test, we compared XCO₂ derived from CO₂ profiles using the MIROC4-ACTM with that where the part of the CO₂ profile above the TROPPB was filled in by extrapolating the aircraft data up to 0.0128 hPa. The difference in XCO₂ was as small as 0.2 ± 0.1 ppm on average.


## 4 Result and Discussion

### 4.1 Spatiotemporal variation of CO₂ seen by ship, aircraft, and satellite

**Figure 3a-c** presents the temporal variation of monthly average CO₂ mixing ratios obtained by ship and aircraft in three representative latitude ranges, namely the northern mid latitudes (20° N–30° N), the equator region (0° N–10° N), and southern

latitudes (20° S–10° S). Ship and aircraft data refer to lower and upper tropospheric CO₂ mixing ratios. The largest seasonal cycle of the CO₂ mixing ratio is seen in the northern hemisphere (NH) at 20° N–30° N. Average CO₂ mixing ratios of 402.9± 3.6 ppm and 401.2 ± 3.1 ppm at lower and upper troposphere, exceeded that from south of the equator by 4.5 ppm and 1.5 ppm, respectively. Maxima occur in April to May at sea-level, which is approximately 1 month earlier than in the upper troposphere (May to June). Minima seen in autumn show a greater temporal variability in the lower troposphere (August to

October) than at about 10 km height (September). At 20° –30° N, the peak-to-trough amplitudes of the seasonal cycles at sea level is 8.5 ± 0.9 ppm, and is ~2 ppm larger than the amplitudes in the upper troposphere (6.5 ± 0.6 ppm). Amplitudes decrease with latitude, showing similar values of about 4 ppm at the equator. In the southern hemisphere (SH), the amplitudes approach 0 at sea level **(Fig. 3c)**. In contrast, the upper troposphere shows two small peaks, one in June and one in November/December in 2014 and 2015, and additionally in April 2016. Seasonal cycles and decreasing amplitudes from North to South (6 ppm to

3 ppm) are similar to that observed by Matsueda et al. (2008) over the same region between 2005 to 2007 using aircraft based





flask samples. At sea-level, seasonal cycle amplitudes that decrease from about 8 ppm at 20° N–30° N to 3 ppm at the equator were reported by the global sampling network of the National Oceanic and Atmospheric Administration's Climate Monitoring and Diagnostics Laboratory (NOAA/CMDL) (Conway et al., 1994). The current observed characteristics are consistent with the previous long-term studies.

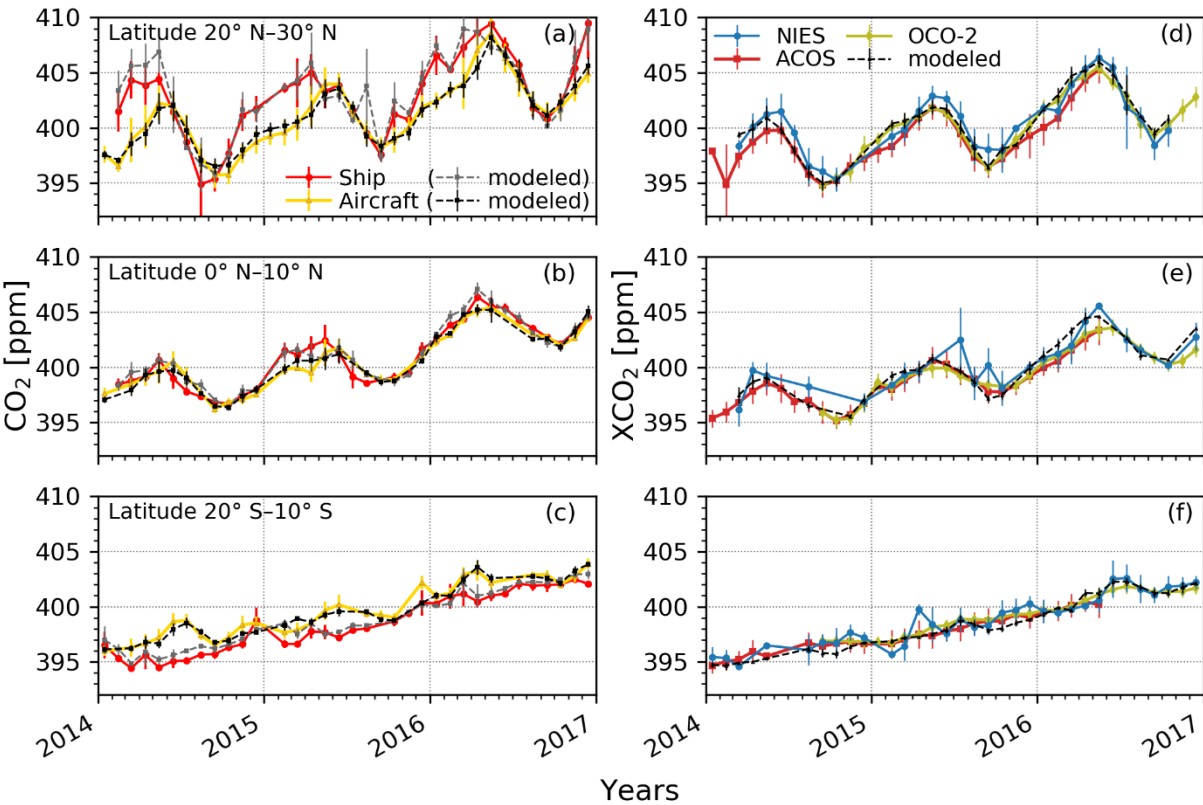


**Figure 3.** Temporal variation of the monthly average $CO_2$ mixing ratio obtained by ship (red) and aircraft (yellow) (left column), and the column averaged mixing ratios ($XCO_2$) from the NIES (blue), ACOS (red), and OCO-2 (olive) (right column) in three representative latitude ranges for the northern mid latitudes a) and d), the equator region b) and e), and southern latitudes c) and f). Results of the ACTM are shown as dashed lines. Error bars represent the standard deviation of the monthly averages.


As the NH transitions from winter to spring, **Fig. 3a** reveals that the $CO_2$ mixing ratio increases rapidly at the surface, but only moderately at the upper troposphere, which results in a difference of up to 4 ppm. In 2014 and 2015, upper tropospheric peak values show a delay of 1 month, which is not seen in 2016, likely due to year-to-year variations. Similar observation have been made previously over the northern Pacific (Miyazaki et al., 2008; Nakazawa et al., 1991) and attributed to the response of the

terrestrial carbon metabolism of the NH (China, Korea, Japan) and predominant northwesterly airmass transport (Umezawa et al., 2018). Specifically, low net primary productivity (NPP) and leaf litter decomposition in autumn to winter is linked to a net carbon release from the terrestrial ecosystem and subsequent increase in the $CO_2$ mixing ratio at the lower troposphere, which





persists until spring. Vertical mixing mitigates the altitude dependent $CO_2$ gradient with a time offset of about 5 months. In spring to summer, high NPP rates substantially removes $CO_2$ from the atmosphere. At that season, strong convection, associated with significant uplift of low-$CO_2$ air masses, results in a well-mixed troposphere (Miyazaki et al., 2008; Nakazawa et al., 1991; Niwa et al., 2011). The flux footprints on upper tropospheric $CO_2$ is generally much wider compared to that near the surface at all latitudes, resulting in a smoother vertical gradients and smaller seasonal cycle amplitudes at higher altitudes.

**Figure 3d-f** presents the temporal variation of column averaged mixing ratios of $CO_2$ (X$CO_2$) retrieved by NIES, ACOS, and OCO-2. The number of valid bias corrected X$CO_2$ retrievals by NIES are less than 25 % of that by ACOS with good quality flag. Seasonal patterns of all retrievals were similar in the NH, showing peaks in late spring/early summer (May to June), and minima in autumn (September to October). While peaks of X$CO_2$ by NIES are higher by 1 to 3 ppm, ACOS and OCO-2 values agree within 1 ppm (**Figs. 3d and 3e**). The largest amplitudes of ACOS and OCO-2 at 20° N–30° N (5 to 6 ppm) are approximately 2 ppm smaller than those of NIES (6 to 8 ppm). Southwards, the strong seasonal cycle decreases, and disappears in the SH, similar to observations made by in situ measurements at sea level. The NIES X$CO_2$ product shows substantial scatter and limited valid data each month at lower latitudes, unlike ACOS and OCO-2 (**Figs. 3e and 3f**). Differences in retrieval algorithms can explain discrepancies in the X$CO_2$ (Reuter et al., 2013), while the reduced number of data points of NIES are likely due to stricter quality filters. The results imply that seasonal variations of $CO_2$ at lower latitudes are better represented by the ACOS/OCO-2 retrieval algorithm.

**Table 1.** Root-mean-square error (RMSE), and average difference and standard deviation between the retrievals from aircraft, ship, satellite and the corresponding results from the ACTM at different latitude ranges between 2014 and 2017.

| | | | RMSE | | |
|---|---|---|---|---|---|
| **Latitude** | **Aircraft** | **Ship** | **NIES** | **ACOS** | **OCO-2** |
| **20° N–30° N** | 0.54 | 1.26 | 0.93 | 1.09 | 0.44 |
| **0° N–10° N** | 0.44 | 0.68 | 1.14 | 0.93 | 0.93 |
| **20° S–10° S** | 0.55 | 0.63 | 0.86 | 0.54 | 0.56 |
| | | **Difference measured in situ or/satellite X$CO_2$ − ACTM (ppm)** | | | |
| **Latitude** | **Aircraft** | **Ship** | **NIES** | **ACOS** | **OCO-2** |
| **20° N–30° N** | 0.00 ± 0.54 | −0.41 ± 1.19 | 0.16 ± 0.92 | −0.81 ± 0.72 | −0.30 ± 0.32 |
| **0° N–10° N** | 0.01 ± 0.44 | −0.20 ± 0.65 | 0.17 ± 1.13 | −0.58 ± 0.72 | −0.51 ± 0.78 |
| **20° S–10° S** | 0.13 ± 0.54 | −0.40 ± 0.48 | 0.33 ± 0.80 | 0.15 ± 0.52 | 0.20 ± 0.52 |

**Figure 3** also presents the simulated X$CO_2$, sea-level $CO_2$ mixing ratios, and upper troposphere $CO_2$ mixing ratios, calculated by the MIROC-4 ACTM. Best agreement is found between the model results in the upper troposphere and the aircraft observations (RMSE 0.51 ± 0.05, average difference 0.05 ± 0.06) (**Table 1**). The largest discrepancy to the model results occur





for the ship observations at northern midlatitudes (RMSE 1.26, difference 0.41 ± 1.19), likely due to the large gradients and variations of $CO_2$ concentrations typically found at this latitude range at sea-level. The coarse horizontal resolution of the model is not adequate to represents observations near source regions. The RMSE of the difference between satellite $XCO_2$ and the MIROC-4 ACTM ranges from 0.44 to 1.14, which may result both from the higher uncertainties of the simulations at sea-level, and the uncertainties in the satellite retrievals. OCO-2 v9r shows systematically higher RMSE around the equator at 0° N–10° N, relative to the 20° N–30° N and 10° S–20° S region.

## 4.2 Latitudinal variations of $CO_2$ seen by ship, aircraft, and satellite

**Figure 4a-c** displays the latitudinal distribution of the $CO_2$ mixing ratio of ship and aircraft for three selected months in 2015, which are representative for different latitudinal $CO_2$ gradients in the troposphere. From North towards the equator, the negative tropospheric $CO_2$ gradient decreases rapidly, especially in spring (March) and autumn (October) (**Figs. 4a and 4c**). Around the equator, ship and aircraft mixing ratios agree within 0.2 ± 0.8 ppm. In the SH, the gradient is reversed, showing upper tropospheric $CO_2$ mixing ratio to be larger by 1.4 ± 0.9 ppm, especially during NH spring to summer (**Fig. 3c, Fig. 4b**). Previous model studies, which included aircraft observations, explain the atmospheric $CO_2$ characteristics south of the equator by meridional transport processes (Miyazaki et al., 2008; Niwa et al., 2011). Our current ACTM forward simulations reveal in particular that $CO_2$, which is strongly emitted during winter to spring (December to May) over NH land, causes a strong meridional $CO_2$ gradient at sea level, and the $CO_2$ rich air is transported towards the equator (**Fig. A2**). In NH summer (June to August), the meridional gradient is substantially weakened due to the seasonal $CO_2$ sink at northern midlatitudes (**Fig. 4b, Fig. A2f-h**). At the upper troposphere, meridional gradients are absent during autumn (September-November) (**Fig. 4c, Fig. A2i-k**) and gradients are weak in winter (December to February) (**Fig. A2l-b**), but increase towards summer due to vertical mixing of $CO_2$ rich air from the surface at northern midlatitudes (**Fig. 4a, Fig. A2c-e**). Near the equator, uplift by convection increase the $CO_2$ mixing ratio in the middle and upper troposphere in all seasons. In the SH, strong meridional transport from the NH to the SH occurs only from late spring to early summer in the upper troposphere during which time the $CO_2$ mixing ratio in the upper troposphere exceeds that at the sea-surface (**Fig. 4b**). Furthermore, $CO_2$ uptake by the Southern Pacific and southern hemispheric land vegetation decrease $CO_2$ at sea-level. The current in situ observations confirm the inter-hemispheric transport mechanism of $CO_2$.



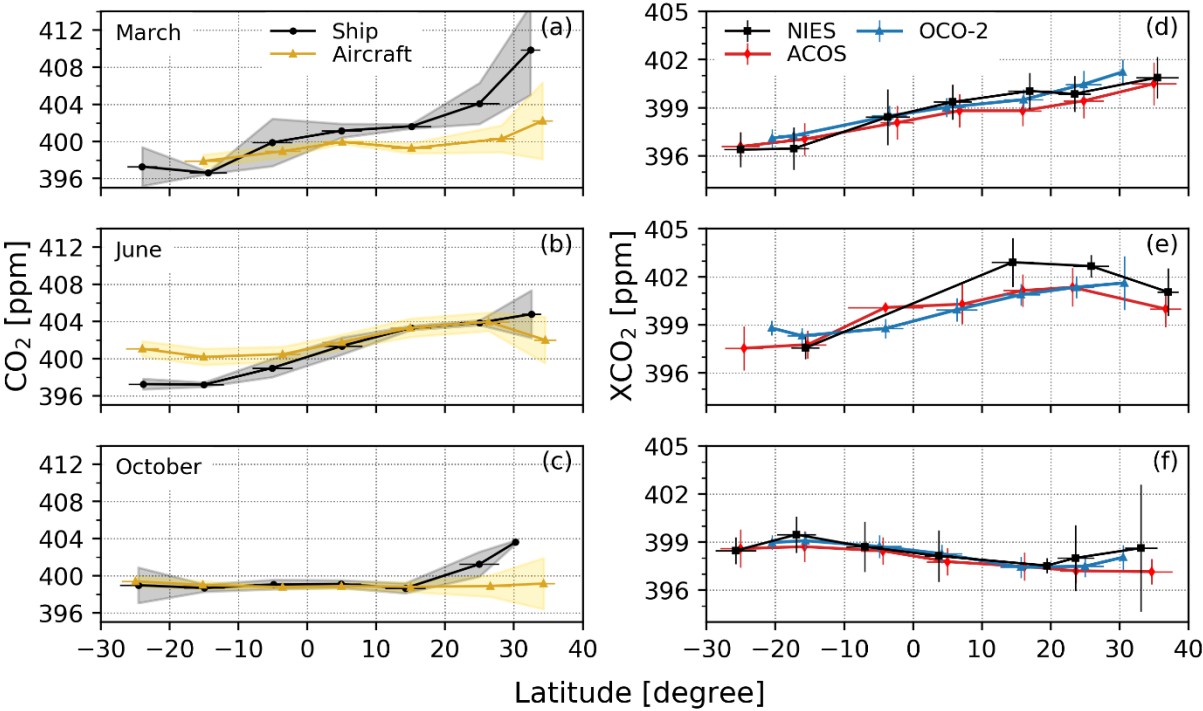

**Figure 4.** Latitudinal distribution of the $CO_2$ mixing ratio obtained by ship (black) and aircraft (yellow) (left column), and of $XCO_2$ obtained
285 by NIES (black), ACOS (red), and OCO-2 (blue) (right column) for three selected months in 2015, which are representative for different
latitudinal $CO_2$ gradients in the troposphere: March a) and d), June b) and e), and October c) and f). Shaded areas are the standard deviation
of the monthly average $CO_2$ mixing ratios. Error bars show the standard deviation of the monthly averaged $XCO_2$, and of the location within
each latitude box.

290 **Figure 4d-f** shows the latitudinal distribution of $XCO_2$ retrieved by NIES, ACOS, and OCO-2. In spring, maximum values

appear in the NH and minima in the SH (**Fig. 4d**). In autumn, the locations of the maxima and minima are reversed between

NH and SH (**Fig. 4f**). In summer (June), the maxima occur at 10° N–20° N (**Fig. 4e**), which is the result of substantial carbon

removal by high NPP at higher latitudes (30° N–40° N) as described above. At that transition point, $XCO_2$ of NIES exceeds

that of ACOS and OCO-2 by about 2 ppm. The in situ and satellite observations reveal the complex $CO_2$ fluxes and transport

295 processes. The results demonstrate that measuring upper and lower tropospheric $CO_2$ mixing ratios simultaneously is important

to better understand $CO_2$ fluxes, which is necessary to further improve atmospheric chemistry transport models. The

consistency of the satellite $XCO_2$ with in situ observations will be evaluated by comparison with the corresponding in situ

$XCO_2$ values in the following section.


### 4.3 Evaluation of seasonal and interannual changes of satellite XCO₂ by combined ship and aircraft observations

**Figure 5a-c** shows the temporal variation of the satellite and in situ derived XCO₂, and the difference between in situ derived and satellite XCO₂ in **Fig. 5d-f**. In all latitudes, in situ and satellite XCO₂ show an overall significant positive correlation ($R^2$: NIES = 0.84 ± 0.02, ACOS = 0.74 ± 0.07, OCO-2 = 0.81 ± 0.04) (**Table 2**). However, in the NH, satellite retrievals are negatively biased by up to 1.6 ± 0.6 ppm (ACOS) at 20° N–30° N (**Fig. 5a and 5d, Table 3**). The smallest bias is found for NIES, likely due to the stricter quality filters as discussed in **section 4.1**. The root-mean-square error (RMSE) of the difference between in situ XCO₂ and satellite XCO₂ is 1.06, 1.26, and 1.70 for NIES, OCO-2, and ACOS respectively, and decreases by 40% (0.53 ppm) on average between the northernmost and southernmost regions (**Table 2**). Agreement within 1 ppm on average is found in the SH (**Fig. 5c and 5f**).

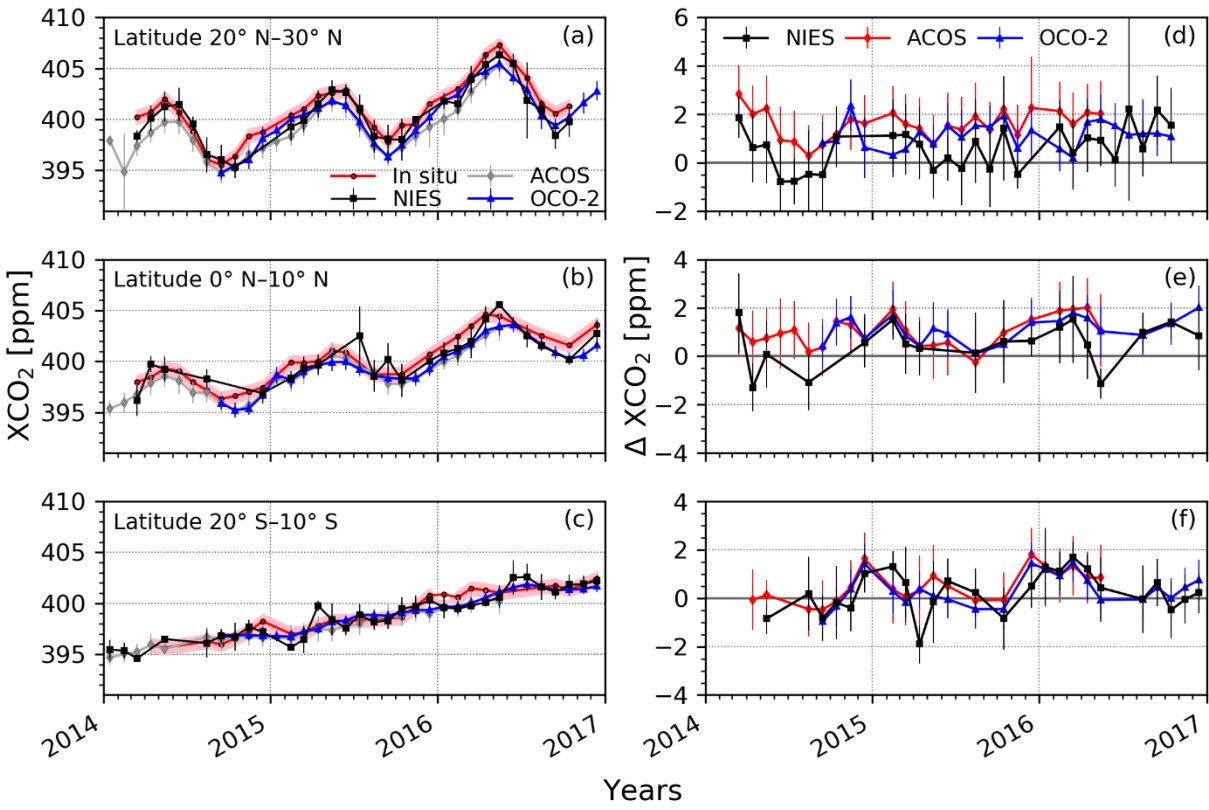

**Figure 5.** Temporal variation of the satellite derived XCO₂ obtained by NIES (black), ACOS (grey), and OCO-2 (blue) in comparison with the in situ XCO₂ (red) (left column), and the difference between in situ derived and NIES (black), ACOS (red), and OCO-2 (blue) (right column) for three selected latitude boxes. Red shaded areas are the uncertainty of the in situ XCO₂ derived from the ±2 ppm variability in the in situ constructed CO₂ profile at ~850 hPa. Error bars show the standard deviation of the monthly averaged XCO₂.





**Table 2.** Coefficient of determination ($R^2$) and root-mean-square error (RMSE) between in situ $XCO_2$ and satellite $XCO_2$ retrievals from GOSAT (NIES, ACOS) and OCO-2 at different latitude ranges between 2014 and 2017.

| Latitude | $R^2$ | | | RMSE | | |
|---|---|---|---|---|---|---|
| | NIES | ACOS | OCO-2 | NIES | ACOS | OCO-2 |
| **20° N–30° N** | 0.86 | 0.64 | 0.81 | 1.06 | 1.70 | 1.26 |
| **0° N–10° N** | 0.81 | 0.76 | 0.76 | 1.02 | 1.17 | 1.23 |
| **20° S–10° S** | 0.84 | 0.81 | 0.87 | 0.87 | 0.83 | 0.73 |

**Table 3.** Average (Avg.) difference and the standard deviation (Std.) between in situ and satellite $XCO_2$ from GOSAT (NIES, ACOS) and OCO-2 of each latitude range between 2014 and 2017.

| Latitude | difference in situ $XCO_2$ – satellite $XCO_2$ | | | | | |
|---|---|---|---|---|---|---|
| | Avg. NIES | Std. | Avg. ACOS | Std. | Avg. OCO-2 | Std. |
| **20° N–30° N** | 0.61 | 0.87 | 1.60 | 0.59 | 1.14 | 0.52 |
| **0° N–10° N** | 0.51 | 0.87 | 1.00 | 0.60 | 1.12 | 0.52 |
| **20° N–10° S** | 0.23 | 0.84 | 0.51 | 0.66 | 0.34 | 0.65 |

**Figure 6** displays the latitudinal gradients and the gradient of the difference between in situ and satellite $XCO_2$ for the three selected months March, June, and October in 2015 as described above (**section 4.2**). It reveals that generally, the largest differences in the NH coincide with the latitude of the monthly $XCO_2$ maxima. Namely, at 30° N–40° N in spring and autumn with up to 3 ppm (between in situ $XCO_2$ and ACOS in March) (**Figs. 6a and 6d**) and in June at 10° N–20° N with a discrepancy of up to 2 ppm (between in situ $XCO_2$ and OCO-2) (**Figs. 6b and 6e**). Atmospheric $CO_2$ mixing ratios in midlatitudes are characterized by high spatiotemporal variability. Therefore, the observed discrepancies in the NH may arise from differences in sample numbers, location and time within each month and latitude-longitude range. In particular, the largest uncertainty in the in situ $XCO_2$ likely results from the constructed $CO_2$ profile in the mid-troposphere, as no observational constraints are available for that part of the atmosphere and simply a linear interpolation between the ship and aircraft data was assumed (**section 3.2**).

However, **Fig. 3a** reveals that ship and aircraft $CO_2$ mixing ratios are very similar in the second half of each year. Model results of the MIROC-4 ACTM confirm vertically uniform $CO_2$ profiles during that period, which lie within the uncertainty range of the in situ constructed profiles (**Fig. A3**). Niwa et al. (2011) found similar straight vertical profiles between June and September in East Asia, based on aircraft observations and model results. Hence, even though no assumption was necessary at that period, the negative bias persists (**Fig. 5d, Fig. 6e**), which indicates that the difference between in situ and satellite $XCO_2$ can be linked to measurement uncertainties of the satellites.



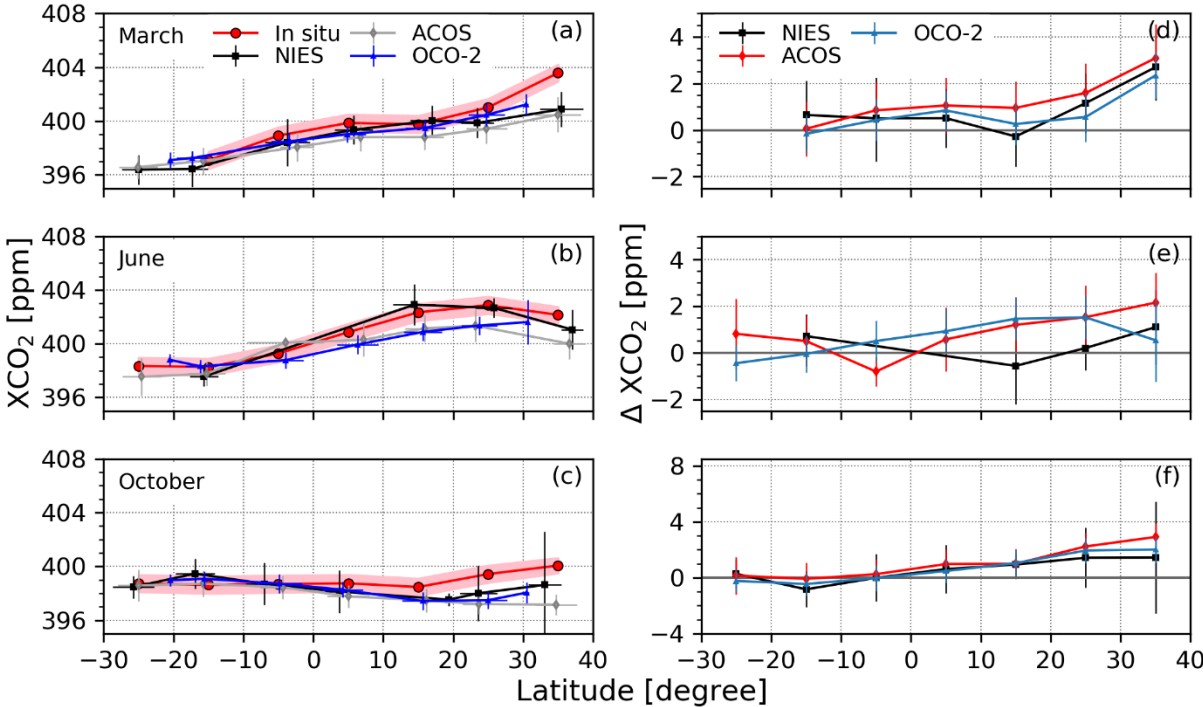

**Figure 6.** Latitudinal gradients of in situ derived $XCO_2$ (red) in comparison with the satellite $XCO_2$ from NIES (black), ACOS (grey), and OCO-2 (blue) (left column), and the difference between the in situ derived $XCO_2$ and NIES (black), ACOS (red), and OCO-2 (blue) (right column) for three selected months, March a) and d), June b) and e), and October c) and f) in 2015. Red shaded areas are the uncertainty of the in situ $XCO_2$ derived from the ±2 ppm variability in the in situ constructed $CO_2$ profile at ~850 hPa. Error bars show the standard deviation of the monthly averaged $XCO_2$ and of the location within each latitude box.

The peak values in the carbon cycle represent the turning points between predominant $CO_2$ sources in boreal winter, and sinks in summer and therefore, are important to constrain changes in the seasonal and interannual variation of the carbon cycle. **Figures 5a and 5b** reveal that maxima and minima generally agree. However, small positive phase shifts of about one month are occasionally observed (2014, 20° N–30° N: maximum of NIES in June; 2014, 10° N–20°N minima of ACOS and OCO-2 in October; 2016, 10° N–20°N: maximum of OCO-2 in June). Long-term measurements (1984 to 2013) observed maxima usually in May and minima in late September in the upper troposphere of the northern West Pacific (Matsueda et al., 2008, 2015). Surface data (between 1987 and 2017) reported maxima in early May and minima in early September over the same region (World Data Centre for Greenhouse Gases (WDCGG) of the World Meteorological Organization (WMO)). The consistency with long-term studies support the correctness of the in situ $XCO_2$, which implies that satellite $XCO_2$ sometimes show a delayed response to $CO_2$ changes.

To explore year-to-year changes in the increase of $XCO_2$, the mean values of the three consecutive highest monthly averages during spring of each year are compared (**Table 4**). Three-month averages around the peak values are chosen due to the limited





data, although usually longer time-periods are needed for that growth calculation. From 2014 to 2015, in situ and satellite $XCO_2$ increased by less than 2 ppm yr$^{-1}$ at 20° N–30° N **(Fig. 5a)**. In contrast, a significant increase of $3.84 \pm 0.65$ ppm yr$^{-1}$

is observed by in situ $XCO_2$ from 2015 to 2016, which is by ~10% larger than that observed by satellites ($3.39 \pm 0.03$). The rapid increase is also seen near the equator, simultaneously with a larger negative bias of the satellite $XCO_2$ in 2016 as compared to the previous years **(Figs. 5b and 5e)**.

**Table 4.** Increase of $XCO_2$ between peaks of consecutive years and the standard error of the difference seen by in situ and satellite $XCO_2$ of GOSAT (NIES, ACOS) and OCO-2 between 2014 and 2017. Peak values are defined as mean of the three consecutive highest monthly averages during spring of each year. In 2016, the mean of ACOS and that of in situ $XCO_2$ at 0° N–10° N is based on 2 months due to limited data. "–" indicates missing data.

| | In situ $XCO_2$ (ppm yr$^{-1}$) | NIES (ppm yr$^{-1}$) | ACOS (ppm yr$^{-1}$) | OCO-2 (ppm yr$^{-1}$) |
|---|---|---|---|---|
| | | **20° N–30° N** | | |
| **2014–2015** | $1.45 \pm 0.63$ | $1.42 \pm 0.60$ | $1.95 \pm 0.54$ | – |
| **2015–2016** | $3.84 \pm 0.65$ | $3.37 \pm 0.43$ | $3.43 \pm 0.40$ | $3.36 \pm 0.38$ |
| | | **0° N–10° N** | | |
| **2014–2015** | $1.72 \pm 0.22$ | – | $1.99 \pm 0.30$ | – |
| **2015–2016** | $3.87 \pm 0.09$ | – | $2.82 \pm 0.37$ | $3.52 \pm 0.16$ |

The larger increase between 2015 and 2016 is likely driven by the strong El Niño in 2015. Matsueda et al. (2008) reported a

mean $CO_2$ growth rate of 1.7 to 1.8 ppm yr$^{-1}$ in 1993 to 2005. However, between 1997 to 1998, they found a significantly enhanced growth rate of about 3 ppm yr$^{-1}$, which they linked to a strong El Niño year (Matsueda et al., 2002, 2008). Indeed, it is well documented that the interannual variation in the growth rate of $CO_2$ is closely linked to the El Niño–Southern Oscillation (ENSO), which affects the carbon cycle though changes in the atmospheric and ocean circulation (e.g., Bacastow, 1976; Keeling, C. D.Revelle, 1985; Kim et al., 2016; Patra et al., 2005; Wang et al., 2013; Zeng et al., 2005). Particularly, the

increase of $CO_2$ was attributed to a decrease in the NPP, increased soil respiration, and enhanced fire emissions related to low precipitation and high temperatures (Liu et al., 2017). Recent model results found that the maximum $CO_2$ growth rate appears several months after the El Niño peak as response to the low NPP (Kim et al., 2016). In fact, the maximum increase observed in this study occurred in NH spring, after the peak of the 2015 El Niño in November/December **(Fig. 5a and 5b)**.

Opposite to the strong increase, in situ $XCO_2$ shows no increase between March and April around the equator in 2015 **(Fig.**

**5b)**. One month earlier (February), a reduction in $XCO_2$ is seen by ACOS and OCO-2. It has been argued that the upwelling of carbon rich water to the surface at the equator is suppressed in the eastern and central Pacific Ocean during El Niño (Feely et al., 2002; Keeling, C. D.Revelle, 1985), which subsequently leads to an initial negative $CO_2$ anomaly over that region (Rayner et al., 1999). Coincident timing of the observed anomalies with different phases of the El Niño suggest that the ocean and terrestrial response to the event affect the atmospheric $CO_2$ mixing ratio even at the study region at 140° E to 160° E.





Supportive to this interpretation, Chatterjee et al. (2017) found a negative anomaly in atmospheric $CO_2$ concentrations over the so-called Niño 3.4 region (120° W–170° W) between March and July 2015 in the OCO-2 retrievals. Consequently, ACOS and OCO-2 reflect the negative anomaly of $CO_2$ of the first phase of the El Niño, whereas in the second phase, the response of the atmospheric $CO_2$ mixing ratio to the event is better represented by the higher growth rate of the in situ $XCO_2$. Given the uncertainties associated with the negative $CO_2$ anomaly observed at the study region, the result therefore suggests that,

compared to satellite observations, in situ $XCO_2$ sometimes show a higher sensitivity to year-to-year changes in the atmospheric $CO_2$ mixing ratio.

## 5 Conclusions

The current study indicates that seasonal, latitudinal and interannual variation of atmospheric $CO_2$ mixing ratios over the open ocean can be accurately determined by in situ derived column average $CO_2$ mixing ratios, defined as in situ $XCO_2$. The

sensitivity of the in situ $XCO_2$ dataset to year-to-year variations was demonstrated on the distinct ocean and terrestrial responses to the 2015–2016 El Niño event around the equator. Namely, a stagnation in the springtime increase during the early stage of the El Niño event was linked to reduced $CO_2$ outgassing from the ocean, and a substantial increase to the later stage, reflecting the increase of $CO_2$ emissions from the terrestrial ecosystem.

The evaluation of three different satellite retrievals (ACOS, NIES, OCO-2) by the in situ $XCO_2$ revealed similar seasonal

pattern ($R^2 = 0.64$–$0.87$). However, a negative bias of $1.12 \pm 0.40$ ppm on average and higher difference in the northern (NH) than in the southern hemisphere (SH) were attributed to measurement uncertainties of the satellites. Compared to ACOS and OCO-2, the NIES retrieval showed higher accuracy in the northern hemispherical midlatitudes. At low latitudes, NIES retrievals show substantial scatter and very few valid data points. ACOS and OCO-2 provide a more reliable analysis of carbon cycles at these latitudes. The seasonal cycle of all retrievals occasionally showed a positive phase shift of one month relative

to the in situ $XCO_2$ at different times of year. In some cases, the representation of year-to-year variations in atmospheric $CO_2$ mixing ratios is more distinct in the in situ $XCO_2$ values as compared to the satellite estimates and therefore, are suggested to be sometimes of higher sensitivity. Hence, the result indicates that even if the retrievals complement each other, measurement uncertainties remain, which limit the accurate interpretation of spatiotemporal changes in $CO_2$ fluxes by satellites alone.

Advanced observations like those from GOSAT-2 and improvements in retrieval algorithms like those from ACOS version 9,

and OCO-2 version 10, increase the number of valid data points at lower latitudes and reduce uncertainties. An initial comparison of the in situ $XCO_2$ dataset with ACOS v9r revealed a decrease of the negative bias by more than 50% at northern midlatitudes as compared to ACOS v7.3 (**Fig. A1**). This example highlights the utility of the in situ $XCO_2$ dataset as a reference for satellite derived $XCO_2$ estimates and to clarify the impacts of changes between different versions of retrieval algorithms.

Our study provides a short-term perspective on the great potential of the new bottom-up approach which can help to understand

changes in the carbon cycle in response to global warming and to interpret their contribution to atmospheric $CO_2$ growth. We





propose that a long-term $XCO_2$ dataset based on co-located $CO_2$ measurements by commercial ships and aircraft can augment TCCON data for validating $XCO_2$ estimates from satellites over the open ocean. To accomplish this objective, these commercial ship and aircraft measurements should be expanded and must be sustained for the foreseeable future.

**Appendix A:**

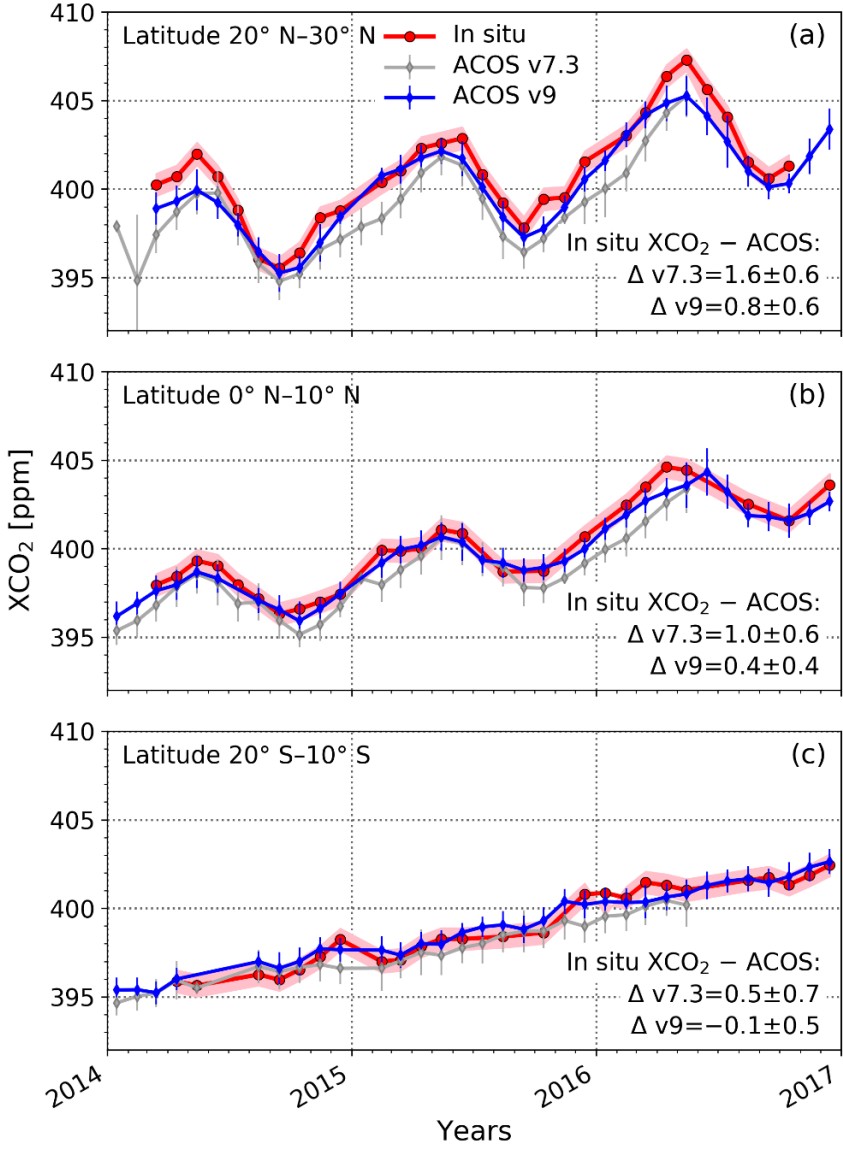


**Figure A1.** Comparison of the temporal variation of in situ $XCO_2$ (red) with $XCO_2$ derived from ACOS v7.3 (grey) and ACOS v9 (blue) for three selected latitude ranges. Red shaded areas are the uncertainty of the in situ $XCO_2$ derived from the ±2 ppm variability in the in situ constructed $CO_2$ profile at ~850 hPa. Error bars show the standard deviation of the monthly averaged $XCO_2$.

**Figure A2**. Latitude-pressure distribution of the inversion of the $CO_2$ mixing ratio at longitude 146° E in 2015, obtained from ACTM forward simulations.



**Figure A3.** In situ constructed $CO_2$ profiles (blue) using ship (SOOP) and aircraft (CONTRAIL) data (yellow), together with the results of the ACTM (green), and the interpolation (red) for the month June and July in 2014 a), b), 2015 c), d), and 2016 e), f) at the latitude range 20° N–30° N.





*Data availability*. The OCO-2 data presented in this manuscript are available from the NASA Goddard GES DISC at
https://disc.gsfc.nasa.gov/datasets/OCO2_L2_Lite_FP_9r/summary (OCO-2 Science Team/Michael Gunson, Annmarie
Eldering, 2018). ACOS data are available at https://disc.gsfc.nasa.gov/datasets/ACOS_L2_Lite_FP_9r/summary (OCO-2
Science      Team/Michael      Gunson,      Annmarie      Eldering,      2019),      and      at
https://disc.gsfc.nasa.gov/datasets/ACOS_L2_Lite_FP_7.3/summary (OCO-2 Science Team/Michael Gunson, Annmarie
Eldering, 2016). GOSAT data are available from the GOSAT Project website of the National Institute for Environmental
Studies ("NIES") at https://data2.gosat.nies.go.jp/index_en.html, accessed: [4/28/2020]. SOOP data are available at
http://soop.jp/, accessed: [9/26/2019]. The CONTRAIL CME $CO_2$ data are available on the Global Environmental Database
of the Center for Global Environmental Studies of NIES (https://doi.org/10.17595/20180208.001). The CONTRAIL data are
also available from the ObsPack data product (http://www.esrl.noaa.gov/gmd/ccgg/obspack/) and the World Data Center for
Greenhouse Gases (https://gaw.kishou.go.jp/).

*Supplementary data.* Supplementary information and data accompany this article.

*Author Contributions*. The study was designed by H.T. Data analyses were made by A.M., and extensive discussions were
made by A.M., H.T., T.S., T.M., P.P., J.L., and D.C. The paper was written, edited, and proofed by all the authors.

*Competing Interests*. The authors declare that they have no competing interests.

*Acknowledgements*. We are grateful to Dr. Naveen Chandra, NIES for providing $CO_2$ inversion fluxes that are used for
simulating atmospheric concentration by ACTM. We also would like to thank the anonymous reviewers for valuable comments
and suggestions improving the paper.
Financial support was given by multiple grants: the Global Environmental Research Coordination System from the Ministry
of the Environment, Japan (E1851, E1253, E1652, E1151, E1951, E1451, E1751, E1432), and the Environmental Research
and Technology Development Fund (ERTDF) from the Ministry of the Environment, Japan (grant no. 2-1803.
JPMEERF20182003).
The authors acknowledge the satellite data infrastructure for providing access to the GOSAT NIES, the GOSAT ACOS and
the OCO-2 data used in this study. This research is a contribution to the Research Announcement on GOSAT series joint
research, entitled "Combined cargo-ship and passenger aircraft observations-based validation of GOSAT-2 GHG observations
over the open oceans", and to the Greenhouse Gas Initiative of the Atmospheric Composition Virtual Constellation (AC-VC)





of the Committee on Earth Observation Satellites (CEOS). We thank the WDCGG (World Data Centre for Greenhouse Gases) for providing $CO_2$ reference data of the Pacific Ocean from the NOAA GMD Carbon Cycle Cooperative Global Air Sampling
Network, 1968-2018 (Principal investigators include Ed Dlugokencky (NOAA)). Part of this work was conducted at the Jet Propulsion Laboratory, California Institute of Technology, under contract to the National Aeronautics and Space Administration. Government sponsorship acknowledged.

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
