# Peer review of "New approach to evaluate satellite derived XCO2 over oceans by integrating ship and aircraft observations"

_Atmospheric Chemistry and Physics, 2020_

## Referee Comment (RC1) · Anonymous Referee #1 · 18 Dec 2020

This paper describes a method of splicing together in situ measurements from ships, from aircraft, and from the ACTM model to create vertical profiles of CO2 over the Pacific Ocean. The vertical profiles are integrated to calculate XCO2 values that are then compared with the OCO-2, ACOS-GOSAT, and NIES-GOSAT retrievals over the same region. It's not clear to me whether ACP is the correct journal for this publication; it seems as though AMT might be a better fit for the paper's stated goals.

General comments:

There are multiple ATom and HIPPO profiles throughout the Pacific – it would very much strengthen this paper if you could find coincident data with HIPPO/ATom profiles

and compare vertical profiles in detail. It would further strengthen the paper if you could extend the most southern box another 4 degrees to 34S, where you could show that the combined in situ + ACTM total column matches that from the (coastal) Wollongong TCCON station (filtering for onshore wind direction, perhaps).

I found the Results and Discussion section confusing in places (see Specific comments for details) and difficult to follow. Uncertainties are large in the differences and trends, and yet conclusions were drawn about whether satellite measurements agreed with the ship+CONTRAIL+ACTM-derived XCO2.

Specific comments:

L38 – Why cite the 2018 value of atmospheric CO2? You could update this using the NOAA value for 2020. L108 – Why do you only use the tropospheric data in your analyses? Wouldn't the lower stratospheric data provide important constraints on the total column and provide a check on the stratospheric model? L125 – "By measuring the amount of light absorbed by CO2 and O2, the column average CO2 dry air mole fraction (XCO2) is estimated by taking ratio of the total column amounts of CO2 and O2, where O2 provides an estimate for the total column of dry air (Wunch et al., 2011)." This is true for TCCON, but I do not believe this is how the ACOS retrievals work. Please clarify. Figure 2 – How does this profile compare with the GINPUT profile? If I understand correctly, the blue stars are a combination of model, in situ, and extrapolated data, is that correct? If so, calling it the "in situ" profile is misleading. L172 – Why use the MIROC-4 ACTM for the stratosphere instead of the GINPUT stratosphere? How do they compare? L335 – "Hence, even though no assumption was necessary at that period, the negative bias persists (Fig. 5d, Fig. 6e), which indicates that the difference between in situ and satellite XCO2 can be linked to measurement uncertainties of the satellites." I do not follow this logic. Why couldn't the bias be caused by a bias in the ACTM stratosphere and not in the satellite retrievals? L353 – "The consistency with long-term studies support the correctness of the in situ XCO2, which implies that satellite XCO2 sometimes show a delayed response to CO2 changes." Again, I do not

follow this argument. The satellites measure the total column in the atmosphere at the time of the measurement. Are you saying that the satellite measurements are wrong? L359 – "In contrast, a significant increase of $3.84 \pm 0.65$ ppm yr$-1$ is observed by in situ XCO2 from 2015 to 2016, which is by $\sim$10% larger than that observed by satellites ($3.39 \pm 0.03$)." Firstly, I don't see $3.39 \pm 0.03$ in Table 4 – is this a typo? Secondly, these numbers do not differ by 10% - their uncertainties overlap and therefore you cannot say anything conclusive about how they differ.

Technical comments:

L55 – change "improves" to "improve" L56 – change "the second NASA" to "NASA's" L71 – TCCON has a very limited number of sites observing *the atmosphere over* open oceans. I'm not sure how you define this, since there are several coastal and island TCCON stations (e.g., Réunion Island, Ascension Island, Izaña, Burgos, Darwin, Wollongong) and the TCCON footprint is large enough that it would be sensitive to CO2 over oceans.
* * *

---

## Author Comment (AC1) · 26 Jan 2021

**General.**

We would like to thank the anonymous Referee #1 for providing comments to improve and clarify our manuscript. We will revise the text by fully taking the comments into account. Please find our responses to the specific comments and questions below.

**5 Comments of Referee #1 and our responses to them**

**Comment**

This paper describes a method of splicing together in situ measurements from ships, from aircraft, and from the ACTM model to create vertical profiles of CO2 over the Pacific Ocean. The vertical profiles are integrated to calculate XCO2 values that are then compared with the OCO-2, ACOS-GOSAT, and NIES-GOSAT retrievals over the same region. It's not clear to me

10 whether ACP is the correct journal for this publication; it seems as though AMT might be a better fit for the paper's stated goals.

**Response**

Our manuscript, which describes a method to derive XCO2 by using ship, aircraft and model data, doesn't intend to solely focus on the technical and theoretical aspects (with a rigorous uncertainty analysis). In addition to the technical aspects, our manuscript presents a detailed analysis of the spatiotemporal variations of CO2 of each in situ and satellite

15 dataset over the Pacific Ocean (section 4.1 and 4.2). Furthermore, using the new constructed in situ XCO2 dataset, we demonstrate its application as reference for XCO2, which is not only of relevance for validating satellite data, but especially for carbon cycle studies. As a complement to TCCON data, we believe that the applicability as reference for XCO2 over oceans is of immediate relevance to a wide interdisciplinary scientific audience in atmospheric chemistry and physical sciences. Because our goal is beyond the primarily technical aspects, we think ACP would fit our goals better.

**General comments:**

• There are multiple ATom and HIPPO profiles throughout the Pacific – it would very much strengthen this paper if you could find coincident data with HIPPO/ATom profiles and compare vertical profiles in detail.

We fully agree with the Referee #1 that HIPPO (Hiaper Pole-to-Pole Observations) and ATom (Atmospheric Tomography Mission) profiles would be very valuable to strengthen our results. However, coincident profile data of the HIPPO and ATom campaigns between the years 2014 and 2017 and in the longitude–latitude range of 130° E to 173° E and 30° S to 40°N do not exist. The newest dataset of HIPPO covers the year 2011 (HIPPO 4, HIPPO 5,

30 https://www.eol.ucar.edu/node/3402, 12/21/2020). Data of the campaign ATom 1 cover the time period from 07/29/2016 to 08/23/2016 (https://daac.ornl.gov/cgi-bin/dsviewer.pl?ds\_id=1581, 12/21/2020). Unfortunately, the flight tracks closest to our study region are generally more than 20° East (Figure R1, purple line). The lack of coincident data is a drawback in strengthening our results, but emphasises the need to expand the amount of reference data.

**Figure R1.** Comparison of the flight track of the Atom 1 campaign with the location of monthly averaged data of CO2 from aircraft (CONTRAIL, green triangle), ship (Trans Future 5 - TF5, blue squares), the satellite retrievals from NIES (yellow diamonds), ACOS (red circles), and OCO-2 (black stars) between 2014 and 2017. Selected regions for the study within 10° latitude by 20° longitude boxes are shown in red frames. Administrative boundaries © EuroGeographics.

40

• It would further strengthen the paper if you could extend the most southern box another 4 degrees to 34S, where you could show that the combined in situ + ACTM total column matches that from the (coastal) Wollongong TCCON station (filtering for onshore wind direction, perhaps).

**45 Response**

50

We agree that this would be beneficial and of wide interest. Unfortunately, south of the latitude 28° S, the aircraft data of CONTRAIL between Narita, Japan, and Sydney, Australia, are only obtained over land (Figure R2, green triangle). Hence, an overlap with ship data over the ocean area is not given. By using our methodology and combining ship data from the open ocean area with aircraft data over land, no realistic CO2 profiles can be obtained. Therefore, we cannot extent the study area to 34° S at present.

**Figure R2.** Location of the CO2 data from aircraft (CONTRAIL, green triangle) and ship (Trans Future 5 - TF5, blue squares) between 2014 and 2017. Selected region for the study within 10° latitude by 20° longitude boxes is shown in the red frame. Administrative boundaries © EuroGeographics.

• I found the Results and Discussion section confusing in places (see Specific comments for details) and difficult to follow.

**60 Response**

55

We will revise the Results and Discussion section to clarify our statements. Please find our replies to the specific comments below.

• Uncertainties are large in the differences and trends, and yet conclusions were drawn about whether satellite measurements agreed with the ship+CONTRAIL+ACTM-derived XCO2.

**Response**

75

We agree that the uncertainties of the differences between the in situ derived  $XCO_2$  and satellite  $XCO_2$  shown in Table 3 and Fig. 5d-f are large, but significant in northern midlatitudes for ACOS (two-sided t-test, significance level  $\alpha$ =0.05).

70 The difference in the trends is not significant at northern latitudes, but at the equator for ACOS and OCO-2 (two-sided t-test, significance level α=0.05) Table 4.

Although uncertainties are not small, the comparison of the in situ derived XCO2 dataset with satellite retrievals gives important indications on how good the retrievals currently are, and if newly revised retrieval algorithm are improved towards minimizing the difference or not. Figure A1 of Appendix A of the manuscript, as well as Figure R3 below, which shows the in situ derived XCO2 and the data of OCO-2 v9 versus OCO-2 v10, illustrate the applicability of our

new in situ derived XCO2 dataset.

---

## Referee Comment (RC2) · Anonymous Referee #2 · 26 Feb 2021

General comments This manuscript details a new approach for evaluating variations of XCO2 over the Oceans by integrating ship, aircraft and model data to create an "in-situ based XCO2" dataset. This dataset is compared with GOSAT and OCO-2 satellite data to evaluate its capabilities. The paper does have value in its contribution to scientific progress and the scientific qualify of the work is good, however at points I think the paper needs to go into more detail of how and why specific parts of this method were done as I was left with several questions concerning this (as mentioned in my specific comments). The paper discusses biases between their in-situ XCO2 and the satellites, concluding that these can be attributed to measurement uncertainties of the satellite observations. I am left unconvinced by this argument and would like to see more

analysis of other possible uncertainties in their in-situ profile to strengthen this claim (I explain in more detail in my specific comments no.11). The study goes on to look at differences in the seasonality of the satellite data vs the in-situ data, which again I am sceptical about because I was unconvinced that these differences aren't due to inaccuracies in the in-situ based columns. Specific comments 1. Page 3, line 71: I think it is misleading to say that TCCON has a very limited number of sites observing open oceans given that there are multiple sites on small islands in the oceans and multiple coastal sites. Again on line 80 you mention comparison to TCCON data in the tropical Pacific region. I think it would help to be more specific about these TCCON sites which are used and where they are. 2. Page 3, line 72: Can you please clarify if this is an ocean bias or if this bias is for land and ocean combined at these sites. 3. Page 4, lines 102-104: The last part of this sentence doesn't make sense to me "if the standard deviation does not exceed 3 ppm". 4. Page 4, line 108: Please could you explain how you determined the tropopause height that you use as a cut off for the aircraft data. Did you use a static 11 km for all measurements or did you calculate it for each time and location? 5. Page 4, methodology: Why did you settle on monthly resolution? I am interested if instead of comparing with monthly averages it would have been possible to compare any of the in-situ data more directly to satellite overpasses on the same day for example. Were there any cases where you were able to do this or were the ship, aircraft and satellite data never on the same day? 6. Page 6, comments on section 3.2 and Figure 2: Naming this constructed profile "in-situ" when it consists of both in-situ and model data is misleading. 7. Page 6, comments on section 3.2 and Figure 2: Please could you comment in more detail why you chose to extrapolate the ship concentrations up to 850 hpa. 8. Page 6, comments on section 3.2 and Figure 2: You say that you extrapolate the aircraft data down to 380 or 400 hpa; and then say that the tropopause pressure is used as upper limit for the extrapolation. So are you also extrapolating the data upwards to the tropopause or is this only the case when the aircraft data are above the tropopause? 9. Page 6, comments on section 3.2 and Figure 2: The vertical grid which you are interpolating onto is not discussed. It is

clear from Figure 2 that your red interpolation line does not have the vertical resolution of the different components which you are using to make the "in-situ" profile. I know that you have done this to match the profile of the satellite retrievals, but you don't explain this. ACOS GOSAT and OCO-2 soundings have a different pressure profile for every sounding because they are built from the surface pressure which does change between soundings. Did you calculate the "in-situ" XCO2 for each satellite sounding separately and then average the monthly/spatial results. Or did you assume one fixed (or perhaps make your own average) profile for pressure, apriori co2 and the averaging kernel? 10. Page 8, line 215: You say that the values you get are similar to Matsueda et al. 2008. Can you be more specific on how similar they are? You do this afterwards with the Conway et al. 1994 numbers so it would paint a more complete picture if you could also give numbers in this instance. 11. I disagree with your assessment that the mid troposphere is the only uncertainty in the in-situ profile which you need to consider. You also should consider the tropopause height as well. We can see from Figure A3 that the interpolated profile deviates from the model profile at the tropopause in panels a and c. I think your concentrations at the tropopause can't be overlooked as an uncertainty because the ACTM and your in-situ profile are clearly in disagreement over the location of the tropopause in some of these figures. It may be that the satellite data and your in-situ XCO2 are different because your tropopause height is incorrect, which would lead to seasonally dependent biases. How much does your tropopause height vary over all of the measurement times and locations in a month? By how much will this change your XCO2 values and can this account for the differences you see to the satellite data? 12. I would be interested to see how the ACTM XCO2 would look if you interpolated it onto the satellite grid in the same way you have done for your in-situ data, and how this would compare with both the satellite and in-situ XCO2. 13. Page 15, line 354: It is unclear to me how satellite XCO2 can show a delayed response to CO2 changes since it is measuring in real time. Please could you elaborate on this. 14. Page 16, line 359: Please add the uncertainty and be more accurate in your "less than 2ppm" number since you are being very precise with the number you are

comparing it to in the next sentence. 15. Page 16, line 360: 3.84 ± 0.65 and 3.39 ± 0.03 overlap so you can't draw conclusions about their difference. 16. You conclude by saying that datasets such as yours can augment TCCON data. Did you consider applying your method to other aircraft and ground data at TCCON sites to compare with TCCON itself? Technical corrections Page 2, line 45: You say that most sites are in North America and Europe, and some are in East Asia and Oceania. This reads as though there are no sites in the other continents, which I don't think was your intention to say. Please reword this sentence. Page 2, line 55: "improves" should be "improve". Page 3, line 74: "Aircrafts" should be "aircraft". Page 3, line 79: Negative symbol isn't the same type of symbol used in previous cases. Page 3, line 80: Remove journal and doi from citation. Page 8, line 210: Missing "N" in "20° –30° N" whilst in other instances you have used "20° N–30° N". Figure A2: Add an x-label.

---

## Author Response (AR1)

**General**

We would like to thank the anonymous Referees for providing valuable comments to improve and clarify our manuscript. We revised our manuscript to fully accommodate the Referees' comments. Please find our responses (shown in bold) to the specific comments of Referee #1 and #2, and the descriptions of changes (in bold italic) made in the revised manuscript below. Page and line numbers of the described changes refer to the revised manuscript without markings. The marked-up version of the revised manuscript is attached to the end of this file. All changes in the revised manuscript are marked using the function "Track Changes" of Microsoft Word.

**Comments of Referee #1 and our responses to them**

**Comment**

This paper describes a method of splicing together in situ measurements from ships, from aircraft, and from the ACTM model to create vertical profiles of CO2 over the Pacific Ocean. The vertical profiles are integrated to calculate XCO2 values that are then compared with the OCO-2, ACOS-GOSAT, and NIES-GOSAT retrievals over the same region. It's not clear to me whether ACP is the correct journal for this publication; it seems as though AMT might be a better fit for the paper's stated goals.

**Response**

**Our manuscript, which describes a method to derive $XCO_2$ by using ship, aircraft and model data, doesn't intend to solely focus on the technical and theoretical aspects (with a rigorous uncertainty analysis). In addition to the technical aspects, our manuscript presents a detailed analysis of the spatiotemporal variations of $CO_2$ of each in situ and satellite dataset over the Pacific Ocean (section 4.1 and 4.2). Furthermore, using the new constructed observation-based $XCO_2$ dataset, we demonstrate its application as a reference for $XCO_2$, which is not only of relevance for validating satellite data, but especially for carbon cycle studies. As a complement to TCCON data, we believe that the applicability as reference for $XCO_2$ over oceans is of immediate relevance to a wide interdisciplinary scientific audience in atmospheric chemistry and physical sciences. Because our goal is beyond the primarily technical aspects, we believe that ACP is a more appropriate journal than AMT.**

**General comments**

**Comment**

There are multiple ATom and HIPPO profiles throughout the Pacific – it would very much strengthen this paper if you could find coincident data with HIPPO/ATom profiles and compare vertical profiles in detail.

30 **Response**

**We fully agree with the Referee #1 that HIPPO (Hiaper Pole-to-Pole Observations) and ATom (Atmospheric Tomography Mission) profiles would be very valuable to strengthen our results. However, coincident profile data of the HIPPO and ATom campaigns between the years 2014 and 2017 and in the longitude–latitude range of 130° E to 173° E and 30° S to 40° N do not exist. The newest dataset of HIPPO covers the year 2011 (HIPPO 4, HIPPO 5,**
35 **https://www.eol.ucar.edu/node/3402, 12/21/2020). Data from the Atom 1 campaign cover the time period from 07/29/2016 to 08/23/2016 (https://daac.ornl.gov/cgi-bin/dsviewer.pl?ds_id=1581, 12/21/2020). Unfortunately, the flight tracks closest to our study region are generally more than 20° east of it (Figure R1, purple line). The lack of coincident data is a drawback in strengthening our results, but emphasises the need to expand the amount of reference data.**

[Figure]

**Figure R1.** Comparison of the flight track of the Atom 1 campaign with the location of monthly averaged data of $CO_2$ from aircraft (CONTRAIL, green triangle), ship (Trans Future 5 - TF5, blue squares), the satellite retrievals from NIES (yellow diamonds), ACOS (red circles), and OCO-2 (black stars) between 2014 and 2017. Selected regions for the study within 10° latitude by 20° longitude boxes are shown in red frames. Administrative boundaries © EuroGeographics.

**Comment**

It would further strengthen the paper if you could extend the most southern box another 4 degrees to 34S, where you could show that the combined in situ + ACTM total column matches that from the (coastal) Wollongong TCCON station (filtering for onshore wind direction, perhaps).

50   **Response**

**We agree that this would be beneficial and of wide interest. Unfortunately, south of the latitude 28° S, the aircraft data of CONTRAIL between Narita, Japan, and Sydney, Australia, are only obtained over land (Figure R2, green triangle). Hence, an overlap with ship data over the ocean area is not given. By using our methodology and combining ship data from the open ocean area with aircraft data over land, no realistic $CO_2$ profiles can be obtained. Therefore, we cannot**
55   **extent the study area to 34° S at present.**

[Figure]

**Figure R2.** Location of the $CO_2$ data from aircraft (CONTRAIL, green triangle) and ship (Trans Future 5 - TF5, blue squares) between 2014 and 2017. Selected region for the study within 10° latitude by 20° longitude boxes is shown in the red frame. Administrative boundaries ©
60   EuroGeographics.

**Comment**

I found the Results and Discussion section confusing in places (see Specific comments for details) and difficult to follow.

**Response**

65   **We revised the Results and Discussion section to clarify our statements. Please find our replies to the specific comments below.**

**Comment**

Uncertainties are large in the differences and trends, and yet conclusions were drawn about whether satellite measurements
70    agreed with the ship+CONTRAIL+ACTM-derived XCO2.

**Response**

**We agree that the uncertainties of the differences between the observation-based XCO$_2$ (obs. XCO$_2$) and satellite XCO$_2$ shown in Table 3 and Fig. 5d-f of the manuscript are large, but significant in northern midlatitudes for ACOS (two-sided t-test, significance level α=0.05).**

75    **The difference in the trends between the satellite retrievals and the obs. XCO$_2$ is not significant at northern latitudes, but at the equator in case of ACOS and OCO-2 (two-sided t-test, significance level α=0.05, Table 4), they are significant. Although uncertainties are not small, the comparison of the obs. XCO$_2$ dataset with satellite retrievals gives important indications on how good the retrievals currently are, and if newly revised retrieval algorithm are improved towards minimizing the difference or not. Figure A1 of Appendix A of the manuscript, as well as Figure R3 below, which shows**
80    **the obs. XCO$_2$ and the data of OCO-2 v9 versus OCO-2 v10, illustrate the applicability of our new observation-based XCO$_2$ dataset.**

[Figure]

**Figure R3.** Comparison of the temporal variation of obs. $XCO_2$ (red) with $XCO_2$ derived from OCO-2 v9 (grey) and OCO-2 v10 (blue) for three selected latitude ranges. Red shaded areas are the uncertainty of the obs. $XCO_2$ which was obtained from the $\pm 2$ ppm variability in the observation-based $CO_2$ profile at ~850 hPa. Error bars show the standard deviation of the monthly averaged $XCO_2$.

**For illustration, we added Figure R3 as Figure A2 to Appendix A of the revised manuscript as described below. The uncertainties connected with our dataset and the uncertainties of the comparison are clarified in the revised manuscript as follows:**

Page 14, lines 321–323: *...and satellite $XCO_2$ in Figs. 5d-f.* **The uncertainties of the obs. $XCO_2$ dataset are estimated to be $0.62 \pm 0.01$ ppm on average, which is derived from the $\pm 2$ ppm variation in the observation-based $CO_2$ profile at 2 km above sea level (section 3.2).**

Page 14, lines 330–334: *Agreement within 1 ppm on average is found in the SH (Figs. 5c and 5f).* **The uncertainties of the differences between obs. $XCO_2$ and the satellite retrievals are large. However, the comparison indicates whether the results of the current satellite retrievals tend to show a systematic positive or negative offset (ACOS, OCO-2), or rather a random discrepancy. This comparison is of importance for revising the retrieval algorithm in future.**

Page 5, lines 139–141: **Furthermore, OCO-2 version 10 was released.** *An initial comparison between ACOS v7.3 and v9,* **and between OCO-2 v9r and v10** *is included in the* **Appendix A (Figs. A1 and A2)** *and section 5 Conclusions.*

Pages 19–20, lines 446–449: *An initial comparison of the* **obs.** *$XCO_2$ dataset with ACOS v9 revealed a decrease of the negative bias by more than 50%* **on average** *as compared to ACOS v7.3 (Fig. A1)* **, and the comparison with OCO-2 v10, a decrease of the average bias by more than 90% as compared to OCO-2 v9r (Fig. A2).**

**Specific comments**

**Comment**

L38 – Why cite the 2018 value of atmospheric CO2? You could update this using the NOAA value for 2020.

**Response**

**We revised the reference as follows:**

Page 2, lines 36–39: *Since the beginning of the Industrial Era in the 1750s, fossil fuel combustion and other human activities have increased the atmospheric concentration of $CO_2$ from approximately 277 ppm to more than **410** ppm in 20**20** (Dlugokencky, E. and Tans P.: Trends in Atmospheric Carbon Dioxide, NOAA/GML; www.esrl.noaa.gov/gmd/ccgg/trends/, last access: 7 January 2021).*

Page 27, lines 562–563: **Dlugokencky, E. and Tans, P.: Trends in Atmospheric Carbon Dioxide, NOAA/GML; www.esrl.noaa.gov/gmd/ccgg/trends/, last access: 7 January 2021.**

**Comment**

L108 – Why do you only use the tropospheric data in your analyses? Wouldn't the lower stratospheric data provide important constraints on the total column and provide a check on the stratospheric model?

**Response**

**First, the CONTRAIL flights rarely went into the lower stratosphere during our study period. Therefore, we could have filled out the lower part of the stratosphere with aircraft data only occasionally. For our methodology, we think it is better to have a consistently constrained stratosphere rather than using measurement data in a few profiles while most of the remaining profiles use the results of the MIROC4-ACTM only. Furthermore, the variation of $CO_2$ above the tropopause height varies much less than in the troposphere and can be successfully modelled (Figure R4).**

**Second, the aim of our study is not to provide a validation of the MIROC4-ACTM in the stratosphere, which is already one of best validated stratospheric models at present using high altitude balloon-borne measurements of $SF_6$ and $CO_2$-age-of-air (Patra et al., 2018).**

**We clarify the reason for excluding aircraft data of the stratosphere as follows:**

Page 4, lines 111–114: *Only those data which were obtained below the tropopause height during the cruise at around 11 km altitude are used. **To define the tropopause height, we used the blended tropopause pressure (TROPPB), which is explained in detail in section 3.2. Data of the lower stratosphere were only occasionally obtained. We screened out those data in order to have a consistent methodology for constructing $CO_2$ profiles as explained in section 3.2.***

**Comment**

L125 – "By measuring the amount of light absorbed by CO2 and O2, the column average CO2 dry air mole fraction (XCO2) is estimated by taking ratio of the total column amounts of CO2 and O2, where O2 provides an estimate for the total column of dry air (Wunch et al., 2011)." This is true for TCCON, but I do not believe this is how the ACOS retrievals work. Please clarify.

**Response**

**The Referee #1 is correct. Generally, $XCO_2$ quantifies the average mixing ratio of $CO_2$ in a column of dry air extending from the Earth's surface to the top of the atmosphere. It is derived by taking the ratio of the column integrated number densities of $CO_2$ and the total column of dry air. For the satellite retrievals, the total column of dry air is primarily derived from the surface pressure, which is mainly retrieved from the $O_2$ A-band in case of OCO-2, ACOS (O'Dell et al., 2012, 2018), and the GOSAT retrieval from NIES (Yoshida et al., 2011, 2013). Furthermore, the definitions of $XCO_2$ vary in how the dry air column is estimated and in how the vertical weighting is done (Crisp et al., 2012; O'Dell et al., 2012).**

**We revised the sentence as follows:**

Page 5, lines 130–132: *By measuring the amount of light absorbed by $CO_2$ and $O_2$, the column average $CO_2$ dry air mole fraction ($XCO_2$) is estimated by taking ratio of the total column amounts of $CO_2$ and the total column of dry air* (**O'Dell et al., 2012, 2018;** Wunch et al., 2011; **Yoshida et al., 2011, 2013**).

**Comment**

Figure 2 – How does this profile compare with the GINPUT profile?

**Response**

**Figure R4 shows the monthly averaged $CO_2$ profile calculated by ginput version 1.0.6 (purple line) in addition to the result of the MIROC4-ACTM and the observation-based $CO_2$ profile of Figure 2.**

**Especially in the lower troposphere, the ginput profile differs. This is explained by the fact that the TCCON prior do not try to capture the effect of emissions and do not ingest global flux datasets nor any longitudinal dependent behaviour. In contrast, the MIROC4-ACTM uses realistic flux and transport simulations and therefore, the ACTM derived profile is close to that derived from in situ measurements. In the stratosphere, the difference between the ACTM and ginput profile is small as compared to the troposphere.**

[Figure]

**Figure R4.** Comparison of the observation-based $CO_2$ profile (blue) with that obtained from the ACTM (green) and the TCCON a-priori profile of $CO_2$, calculated by ginput version 1.0.6 (purple). The example is obtained at the latitude 20° N–30° N, March 2014.

175 **We clarify why we used the results of the ACTM instead of those from ginput in the response to the Referee's comment L172 below (Lines 207–231 of the response file).**

**Comment**

If I understand correctly, the blue stars are a combination of model, in situ, and extrapolated data, is that correct? If so, calling

180 it the "in situ" profile is misleading.

**Response**

**The Referee #1 is correct, the blue stars are a combination of model, in situ, and extrapolated data. We intended to make it clear by using the terms "in situ constructed" in Figure 2 and "in situ adjusted profile" in the Figure caption. We revised the labelling in Figure 2, as well as in Figure A3 of Appendix A (Figure A4 of the revised manuscript) and**

185 **the figure captions by using the term "observation-based profile". We also clarified the definition of the "observation-based profile" in the figure captions. Furthermore, we revised the naming of the resulting XCO₂ from "in situ XCO₂" to "observation-based XCO₂" (obs. XCO₂) throughout the whole the manuscript. Because the number of the occurrence of "in situ XCO₂" is large, we did not list each of the changes separately. The changes made are as follows:**

[Figure]

190

Page 8, lines 192–193: *Figure 2. Construction of the **observation-based** $CO_2$ profile (blue) **obtained** by using ship (SOOP) and aircraft (CONTRAIL) data (yellow) together with the results of the ACTM (green), and the interpolation (red). The example is obtained at the latitude 20° N–30° N, March 2014.*

[Figure]

195

Page 24, lines 471–473: *Figure A4. **Observation-based** $CO_2$ profiles (blue) **obtained by** using ship (SOOP) and aircraft (CONTRAIL) data (yellow), together with the results of the ACTM (green), and the interpolation (red) for the month June and July in 2014 a), b), 2015 c), d), and 2016 e), f) at the latitude range 20° N–30° N.*

200    Lines 21, 30, 91, 169, 192, 205, 340, 364, 375, 428, 459-460: Changed from *in situ constructed/ in situ derived* to **observation-based.**

**Comment**

205    L172 – Why use the MIROC-4 ACTM for the stratosphere instead of the GINPUT stratosphere? How do they compare?

**Response**

**First, using the MIROC4-ACTM means that our method is fully independent of TCCON, which is important for using our methodology as complement for evaluating satellite retrievals. If TCCON and our method are independent, and satellite retrievals show a similar bias to both datasets, then the observed difference is a bias of the satellite data. In**

210    **addition, the ginput prior is used in the OCO-2 v10. Therefore, having an independent stratosphere is a good cross check.**

**Second, the MIROC4-ACTM simulates the realistic $CO_2$ fluxes and transport processes as described above. Ginput starts from the average $CO_2$ mixing ratio measured at Mauna Loa and American Samoa and then imposes a latitudinally dependent seasonal cycle to derive the specific $CO_2$ profiles. Nevertheless, the $CO_2$ mixing ratios in the**

215    **stratosphere are varying much less than in the troposphere and therefore, the results of the MIROC4-ACTM and ginput are similar as seen in Figure R4. We clarify the choice of the MIROC4-ACTM as follows:**

Page 8, lines 195–199: *To account for the stratospheric partial column, we used results of the MIROC4-ACTM (Patra et al., 2018) above the TROPPB **(Fig. 2)** instead of the results from ginput. First, by using the MIROC4-ACTM, our method is*

220    *fully independent of TCCON, which is important for using our methodology as a complement to TCCON to evaluate satellite retrievals. Second, the MIROC4-ACTM uses realistic flux and transport simulations and is one of the best validated stratospheric models at present.*

Page 9, lines 215–217: *Second, **as mentioned earlier,** the MIROC4-ACTM is among the best validated stratospheric models*

225    *using high altitude balloon-borne measurements of SF₆ and CO₂-age-of-air (Patra et al., 2018)**, and in the upper troposphere and lower stratosphere using CONTRAIL observations (Bisht et al., 2021)*.

Page 26, lines 522–524: ***Bisht, J. S. H., Machida, T., Chandra, N., Tsuboi, K., Patra, P. K., Umezawa, T., Niwa, Y., Sawa, Y., Morimoto, S., Nakazawa, T., Saitoh, N. and Takigawa, M.: Seasonal Variations of SF₆, CO₂, CH₄, and N₂O in the***

230 *UT/LS Region due to Emissions, Transport, and Chemistry, J. Geophys. Res. Atmos., 126(4), 1–18, doi:10.1029/2020JD033541, 2021.*

**Comment**

235 L335 – "Hence, even though no assumption was necessary at that period, the negative bias persists (Fig. 5d, Fig. 6e), which indicates that the difference between in situ and satellite XCO2 can be linked to measurement uncertainties of the satellites." I do not follow this logic. Why couldn't the bias be caused by a bias in the ACTM stratosphere and not in the satellite retrievals?

**Response**

**The variation of $CO_2$ in the stratosphere is much less as in the troposphere and can be simulated with high precision**
240 **by the MIROC4-ACTM (Patra et al., 2018). Furthermore, as described in Lines 218–220 of the revised manuscript, our sensitivity test revealed that the error of the calculated $XCO_2$ due to the stratospheric part is only $0.2 \pm 0.1$ ppm on average. Based on the error induced by the stratosphere and the uncertainty derived from the variability in the observation-based $CO_2$ profile at ~850 hPa ($0.62 \pm 0.01$ ppm), the largest reasonable bias is 0.9 ppm. This bias is not enough to explain the observed average negative discrepancy of $1.2 \pm 0.4$ ppm for ACOS and OCO-2 from June to**
245 **September between 2014 and 2017.**
**We added the impact of the stratosphere as follows:**

Page 16, lines 364–367: *Niwa et al. (2011) found similar straight vertical profiles between June and September in East Asia, based on aircraft observations and model results.* ***Furthermore, the maximum bias due to errors in the MIROC4-ACTM***
250 ***stratospheric $CO_2$ profile (0.9 ppm) is smaller than the average difference of $1.2 \pm 0.4$ ppm between the obs. $XCO_2$ and satellite observations of ACOS and OCO-2 between June and September (section 3.2).***

**Comment**

L353 – "The consistency with long-term studies support the correctness of the in situ XCO2, which implies that satellite XCO2
255 sometimes show a delayed response to CO2 changes." Again, I do not follow this argument. The satellites measure the total column in the atmosphere at the time of the measurement. Are you saying that the satellite measurements are wrong?

**Response**

**As described in Lines 382–385 of the revised manuscript, long-term in situ measurements in the upper troposphere and at surface level report maxima and minima of $CO_2$ not later than May and September, while the satellite retrievals**
260 **sometimes show the extreme values one month later. Based on the long-term in situ datasets, maxima in June and**

**minima in October are too late. We do not intend to say that the satellite measurements are wrong. Our observations suggest that these positive phase shifts of the satellite data are caused by remaining uncertainties which are introduced by limitations in the retrieval algorithm or the lack of validation data and therefore, have not been previously identified. The lack of validation data makes it difficult to characterize and correct these uncertainties. We know that from**

265 **GOSAT and OCO-2 the retrieval algorithm to obtain $XCO_2$ from the measured radiance are undergoing rapid progress with almost one new version per year for OCO-2. We are hoping that the highly accurate ship and aircraft data over a unique geographical region will help us to build the capacity for the validation of satellite $XCO_2$ retrievals.**
**We clarify our statement as follows:**

270 Page 17, lines 385–388: *The consistency with long-term studies support the correctness of the **obs.** $XCO_2$, which implies that satellite $XCO_2$ sometimes show a delayed response to $CO_2$ changes**, which might be caused by remaining uncertainties introduced by limitations in the retrieval algorithms and have not been previously identified due to the lack of validation data over the open ocean***

275 **Comment**

L359 – "In contrast, a significant increase of 3.84 ± 0.65 ppm yr−1 is observed by in situ XCO2 from 2015 to 2016, which is by _10% larger than that observed by satellites (3.39 ± 0.03)." Firstly, I don't see 3.39 ± 0.03 in Table 4 – is this a typo? Secondly, these numbers do not differ by 10% - their uncertainties overlap and therefore you cannot say anything conclusive about how they differ.

280 **Response**

**On average, the increase of the mean values of all three satellite retrievals is 3.39 ± 0.03 ppm. That means, the value 3.39 ± 0.03 ppm is the average increase and its standard deviation of the mean values of all satellite retrievals in the period 2015 to 2016. We added a column to Table 4 with the average values as shown below.**

**The Referee #1 is correct that the uncertainties overlap. The difference in the increase of the obs. $XCO_2$ and that of the**

285 **satellite $XCO_2$ is not significant at northern latitudes, but the increase of the obs. $XCO_2$ tends to be slightly higher. At the equator, the increase of the obs. $XCO_2$ is significantly higher than that of ACOS and OCO-2 (two-sided t-test, significance level α=0.05). We revised this part as follows:**

Page 18, lines 392–396: *In contrast, a significant increase of 3.84 ± 0.65 ppm $yr^{-1}$ is observed by **obs.** $XCO_2$ from 2015 to*

290 *2016. **The average increase of the mean values of all satellite retrievals is 3.39 ± 0.03 ppm $yr^{-1}$.** This rapid increase is also seen near the equator, **where the increase of the obs. $XCO_2$ is significantly higher than that of ACOS and OCO-2 (two-sided***

*t-test, significance level α=0.05).* **S***imultaneously,* *a larger negative bias of the satellite XCO$_2$ in 2016 as compared to the previous years* **is observed** *(Figs. 5b and 5e).*

295    Page 18, lines 398–401: **Table 4.** *Increase of XCO$_2$ between peaks of consecutive years and the standard error of the difference seen by* **obs.** *and satellite XCO$_2$ of GOSAT (NIES, ACOS) and OCO-2 between 2014 and 2017. Peak values are defined as mean of the three consecutive highest monthly averages during spring of each year. In 2016, the mean of ACOS and that of in situ XCO$_2$ at 0° N–10° N is based on 2 months due to limited data. "–" indicates missing data.* **The right column shows the average increase of all satellite means and its standard deviation.**

300

| | **Obs. XCO$_2$** (ppm yr$^{-1}$) | NIES (ppm yr$^{-1}$) | ACOS (ppm yr$^{-1}$) | OCO-2 (ppm yr$^{-1}$) | **Avg. all satellites** (ppm yr$^{-1}$) |
|---|---|---|---|---|---|
| | | | 20° N–30° N | | |
| 2014–2015 | 1.45 ± 0.63 | 1.42 ± 0.60 | 1.95 ± 0.54 | – | **1.68 ± 0.26** |
| 2015–2016 | 3.84 ± 0.65 | 3.37 ± 0.43 | 3.43 ± 0.40 | 3.36 ± 0.38 | **3.39 ± 0.03** |
| | | | 0° N–10° N | | |
| 2014–2015 | 1.72 ± 0.22 | – | 1.99 ± 0.30 | – | **–** |
| 2015–2016 | 3.87 ± 0.09 | – | 2.82 ± 0.37 | 3.52 ± 0.16 | **3.17 ± 0.35** |

**Technical comments**

**Comment**

305    L55 – change "improves" to "improve"

**Response**

**We revised the sentence as follows:**

Page 2, lines 55–57: *These observations are most sensitive to the lower troposphere where CO$_2$ is most variable (Patra et al., 2003) and therefore, are able to improve the knowledge on local CO$_2$ emission and sinks (Connor et al., 2008).*

310

**Comment**

L56 – change "the second NASA" to "NASA's"

**Response**

**We revised the sentence as follows:**

315     Page 2, lines 58–60: *Japan's Greenhouse gases Observing Satellite (GOSAT), and the second NASA's (National Aeronautics and Space Administration) Orbiting Carbon Observatory (OCO-2) are dedicated to inferring the concentration of GHGs from high-resolution spectra at NIR and SWIR wavelengths.*

**Comment**

320     L71 – TCCON has a very limited number of sites observing \*the atmosphere over\*open oceans. I'm not sure how you define this, since there are several coastal and island TCCON stations (e.g., Réunion Island, Ascension Island, Izaña, Burgos, Darwin, Wollongong) and the TCCON footprint is large enough that it would be sensitive to CO2 over oceans.

**Response**

**The Referee #1 is correct that there are TCCON stations at coastal and island sites. However, wide areas over the open**
325     **ocean, which we define as the area outside the coastal region, are not covered by TCCON stations as shown in Figure R5. Therefore, we speak of "very limited number". Even though the footprints of TCCON stations are large, reference data measured directly above open water areas provide a valuable complement to TCCON stations.**

[Figure]

330     **Figure R5.** Location of current, future, and previous TCCON stations. Data are from 2/11/2020 (https://tccon-wiki.caltech.edu/Main/TCCONSites, 1/8/2021).

**We clarified the definition of "the atmosphere over the open ocean" as follows:**

335     Page 3, lines 73–74: However, TCCON **sites are land based and** very limited number of sites observ**e the atmosphere over** open oceans**, which are defined as the ocean area outside the coastal region**.

**Comments of Referee #2 and our responses to them**

340 **General comments**

**Comment**

This manuscript details a new approach for evaluating variations of XCO2 over the Oceans by integrating ship, aircraft and model data to create an "in-situ based XCO2" dataset. This dataset is compared with GOSAT and OCO-2 satellite data to evaluate its capabilities. The paper does have value in its contribution to scientific progress and the scientific qualify of the

345 work is good, however at points I think the paper needs to go into more detail of how and why specific parts of this method were done as I was left with several questions concerning this (as mentioned in my specific comments). The paper discusses biases between their in-situ XCO2 and the satellites, concluding that these can be attributed to measurement uncertainties of the satellite observations. I am left unconvinced by this argument and would like to see more analysis of other possible uncertainties in their in-situ profile to strengthen this claim (I explain in more detail in my specific comments no.11). The

350 study goes on to look at differences in the seasonality of the satellite data vs the in-situ data, which again I am sceptical about because I was unconvinced that these differences aren't due to inaccuracies in the in-situ based columns.

**Response**

**In the revised manuscript, we go into more detail of how and why we did specific parts of our methodology so that open questions are clarified. We also extended our analysis regarding the uncertainties of the observation-based profiles and**

355 **considered these additional uncertainties for the comparison between observation-based XCO$_2$ (obs. XCO$_2$) and the satellite data. The details are explained under the specific comments below.**

**Specific comments**

360 **Comment**

1. Page 3, line 71: I think it is misleading to say that TCCON has a very limited number of sites observing open oceans given that there are multiple sites on small islands in the oceans and multiple coastal sites. Again on line 80 you mention comparison to TCCON data in the tropical Pacific region. I think it would help to be more specific about these TCCON sites which are used and where they are.

365 **Response**

**We agree with the Referee #2 that there are several coastal and island TCCON stations like Réunion Island, Ascension Island, Izaña, Burgos, Darwin, or Wollongong. This was also pointed out by Referee #1. As we replied to Referee #1, we wanted to emphasize that wide areas over the open ocean are not covered by TCCON stations. In the revised**

manuscript, we define "open ocean" as the area outside the coastal region. We understand that the footprints of TCCON stations are large, but we think that reference data measured directly above open water areas provide a valuable complement to TCCON stations. Regarding the TCCON site mentioned in line 80, we refer to the station in Burgos at 19° N. Against XCO₂ data of this station, data of OCO-2 v8 have a bias of −0.7 ppm (Kulawik et al., 2019). We clarified these points in the revised manuscript as follows:

Page 3, lines 73–74: *However, TCCON* **sites are land based and** *very limited number of sites observe* **the atmosphere over** *open oceans***, which are defined as the ocean area outside the coastal region**.

Page 3, lines 81–83: *More recent comparisons of OCO-2 XCO₂ estimates to in situ measurements from the NASA Atmospheric Tomography Mission reveals a systematic bias of −0.7 ppm over the tropical Pacific, that is also seen in* **the** *data* **at Burgos, a TCCON station** *in that region* (**Kulawik et al., 2019; Velazco et al., 2017**)*.*

Page 28, lines 598–602: ***Kulawik, S. S., Crowell, S., Baker, D., Liu, J., Mckain, K., Sweeney, C., Biraud, S. C., Wofsy, S., Dell, C. W. O., Wennberg, P. O., Wunch, D., Roehl, M., Deutscher, N. M., Kiel, M., Griffith, D. W. T., Velazco, V. A., Dubey, M. K., Sepulveda, E., Elena, O., Rodriguez, G., Té, Y., Heikkinen, P., Dlugokencky, E. J., Gunson, M. R., Eldering, A., Fisher, B. and Osterman, G. B.: Characterization of OCO-2 and ACOS-GOSAT biases and errors for CO₂ flux estimates, (October), 2019.***

Page 30, lines 678–680: ***Velazco, V. A., Morino, I., Uchino, O., Hori, A., Kiel, M., Bukosa, B., Deutscher, N. M., Sakai, T., Nagai, T., Bagtasa, G., Izumi, T., Yoshida, Y. and Griffith, D. W. T.: TCCON Philippines: First measurement results, satellite data and model comparisons in Southeast Asia, Remote Sens., 9(12), 1–18, doi:10.3390/rs9121228, 2017.***

**Comment**

2. Page 3, line 72: Can you please clarify if this is an ocean bias or if this bias is for land and ocean combined at these sites.

**Response**

**It is an ocean bias. It is the average bias between GOSAT ocean H-Gain soundings and the TCCON stations Bremen, Saga, Burgos, Darwin, and Reunion. We clarified our statement as follows:**

Page 3, lines 74–75: *Between the GOSAT NIES* **soundings over the ocean** *and TCCON sites near the ocean, a bias of −1.09 ± 2.27 ppm was found (Morino et al., 2020).*

**Comment**

3. Page 4, lines 102-104: The last part of this sentence doesn't make sense to me "if the standard deviation does not exceed 3 ppm".

**Response**

405  **We clarified the sentence as follows:**

Page 4, lines 105– 108: *Forty seconds (s) after the switch from standard gas to air sample, data are collected as averages of 10 s during the ascent and decent, and 1 min averages during the cruise (~ 15 km horizontal distance).* ***Data of each 10 s and 1 min period are rejected*** *if the standard deviation exceeds 3 ppm (Umezawa et al., 2018).*

410

**Comment**

4. Page 4, line 108: Please could you explain how you determined the tropopause height that you use as a cut off for the aircraft data. Did you use a static 11 km for all measurements or did you calculate it for each time and location?

**Response**

415  **We used the blended tropopause pressure (TROPPB) as cut off for the aircraft data. The TROPPB was extracted from GEOS-FP (Goddard Earth Observing System – forward processing) meteorology data using the python suite "ginput" version 1.0.6 (Laughner et al., 2021). It was calculated for every 3 hours (00, 03, 06, 09, 12, 15, 18, 21 o'clock UTC) for the 5th, 15th, and 25th of each month at each centre location of the 10° latitude by 20° longitude grid.**

420  **We added the explanation on how we determined the tropopause height as follows:**

Page 4, lines 111–114: *Only those data which were obtained below the tropopause height during the cruise at around 11 km altitude are used.* ***To define the tropopause height, we used the blended tropopause pressure (TROPPB), which is explained in detail in section 3.2. Data of the lower stratosphere were only occasionally obtained. We screened out those data in order***

425  ***to have a consistent methodology for constructing $CO_2$ profiles as explained in section 3.2.***

Page 7, lines 181–188: ***The TROPPB*** *is defined as a combination of a thermal tropopause- and dynamic tropopause pressure (Wilcox et al., 2012). The TROPPB data are extracted from GEOS-FP (Goddard Earth Observing System – forward processing) meteorology data using the python suite "ginput"* ***version 1.0.6*** **(Laughner et al., 2021)**. *Ginput was developed*

430  *to generate a priori vertical mixing ratios of chemical species (e.g., $CO_2$, $CO$, $CH_4$, $N_2O$) for the open source TCCON retrieval algorithm, GGG2020 (Laughner et al., in prep).* ***The TROPPB was calculated every 3 hours on the 5th, 15th, and 25th of each***

*month for each centre location of the 10° latitude by 20° longitude boxes. Between 2014 and 2017, the highest monthly variation was found at 20° N–30° N with a standard deviation ranging from 0 to 24 hPa (0.02 to 23.77 hPa) and an average standard deviation of 10 ± 5 hPa. The maximum difference of 24 hPa at the level of the TROPPB corresponds to difference*

435 *in the altitude of 1 to 2 km.*

Page 28, lines 606–607: *Laughner, J., Andrews, A., Roche, S., Kiel, M. and Toon, G.: ginput v1.0.7b: GGG2020 prior profile software, CaltechDATA., 2021.*

440 **Comment**

5. Page 4, methodology: Why did you settle on monthly resolution? I am interested if instead of comparing with monthly averages it would have been possible to compare any of the in-situ data more directly to satellite overpasses on the same day for example. Were there any cases where you were able to do this or were the ship, aircraft and satellite data never on the same day?

445 **Response**

**There were daily coincidences of satellite, aircraft, and ship data. However, the number was limited. Because our primary focus was to analyse the seasonal, interannual and latitudinal variation of $XCO_2$, we decided to settle on monthly resolution in the current study.**

**We clarified why we used a monthly resolution in the revised manuscript as follows:**

450

Page 6, lines 160–161: *For the analysis of the seasonal and interannual variation of $CO_2$, we chose the* monthly averages of the satellite, in situ, and model datasets.

**Comment**

455 6. Page 6, comments on section 3.2 and Figure 2: Naming this constructed profile "in-situ" when it consists of both in-situ and model data is misleading.

**Response**

**We agree that the naming is misleading. Referee #1 criticised the same point. We revised the labelling in Figure 2, as well as in Figure A3 of Appendix A (Figure A4 of the revised manuscript) and the figure captions by using the term**

460 **"observation-based" profile. We also clarified the definition of the "observation-based profile" in the figure captions. Furthermore, we revised the naming of the resulting $XCO_2$ from "in situ $XCO_2$" to "observation-based $XCO_2$" (obs.**

**XCO₂) throughout the whole manuscript. Because the number of the occurrence of "in situ XCO₂" is large, we did not list each of the changes separately. The changes made are as follows:**

[Figure]

465

Page 8, lines 192–193: *Figure 2. Construction of the* **observation-based** *CO₂ profile (blue)* **obtained** *by using ship (SOOP) and aircraft (CONTRAIL) data (yellow) together with the results of the ACTM (green), and the interpolation (red). The example is obtained at the latitude 20° N–30° N, March 2014.*

470    Lines 21, 30, 91, 169, 192, 205, 340, 364, 375, 428, 459-460: Changed from *in situ constructed/ in situ derived* to **observation-based.**

**Comment**

7. Page 6, comments on section 3.2 and Figure 2: Please could you comment in more detail why you chose to extrapolate the
475    ship concentrations up to 850 hpa.

**Response**

**We assumed that the impact of the CO₂ measured at sea level reaches up to about 850 hPa. This corresponds to the 3ʳᵈ and 4ᵗʰ pressure level of the GOSAT NIES and ACOS retrieval, counted from the surface. It represents the boundary layer in which most of the rather rapid changes in the CO₂ mixing ratio occur according to previous studies of**
480    **Frankenberg et al. (2016) and Inai et al. (2018). At pressure levels above, the CO₂ mixing ratios are rather stable or keep changing linearly up to about the tropopause height (Frankenberg et al., 2016; Inai et al., 2018).**
**We clarified why we extrapolate the ship data up to ~850 hPa as follows:**

Pages 6–7 , lines 171–177: *Ship data are extrapolated vertically to ~850 hPa, **which corresponds to the 3$^{rd}$ and 4$^{th}$ pressure level of NIES and ACOS, respectively, counted from the surface**. **We chose this cut off as it represents the boundary layer above sea level in which most of the rapid variation of $CO_2$ occur**. Previous balloon and aircraft measurements by the HIPPO campaign over the Tropical Eastern and Western Pacific showed **stronger** $CO_2$ variation of **about** 1 to 2 ppm within the first 2 km above sea level. **Above this level, the $CO_2$ mixing ratios were rather stable or kept changing linearly up to about the tropopause height** (Frankenberg et al., 2016; Inai et al., 2018). To account for that variation **within the boundary layer**, we added ±2 ppm uncertainty to the $CO_2$ estimates **at ~850 hPa**.*

**Comment**

8. Page 6, comments on section 3.2 and Figure 2: You say that you extrapolate the aircraft data down to 380 or 400 hpa; and then say that the tropopause pressure is used as upper limit for the extrapolation. So are you also extrapolating the data upwards to the tropopause or is this only the case when the aircraft data are above the tropopause?

**Response**

**We are also extrapolating the aircraft data upwards to the height of the blended tropopause pressure. Aircraft data above the tropopause were excluded before the analysis in order to have a consistent stratosphere as explained in our response to Referee #1 in Lines 126–129 of the response file. Similarly, Ohyama et al. (2020) extrapolated profile data obtained by an aircraft upwards to the tropopause height at the TCCON site Burgos.**
**We clarified our explanation as follows:**

Page 7, lines 177–180: *Aircraft data from the cruise portion of the flight, which is usually between 380 and 200 hPa, are selected. These aircraft data are extrapolated down to the lower cruising height limit at 380 hPa, and at 30° N–40° N at 400 hPa. **Furthermore, the aircraft data is also extrapolated upwards to the** blended tropopause pressure (TROPPB).*

**Comment**

9. Page 6, comments on section 3.2 and Figure 2: The vertical grid which you are interpolating onto is not discussed. It is clear from Figure 2 that your red interpolation line does not have the vertical resolution of the different components which you are using to make the "in-situ" profile. I know that you have done this to match the profile of the satellite retrievals, but you don't explain this. ACOS GOSAT and OCO-2 soundings have a different pressure profile for every sounding because they are built from the surface pressure which does change between soundings. Did you calculate the "in-situ" XCO2 for each satellite sounding separately and then average the monthly/spatial results. Or did you assume one fixed (or perhaps make your own average) profile for pressure, apriori co2 and the averaging kernel?

515   **Response**

**Thank you for pointing out our missing explanation of the grid used for the interpolation. In order to compare the calculated XCO₂ based on the observation-based profile with that from the satellite soundings, we did the following. We interpolated the observation-based profiles based on the monthly averaged data on the corresponding monthly averaged pressure grid of the ACOS retrieval for the time period 2014 to May 2016. After that date, we used the**

520   **monthly averaged pressure grid of NIES due to the temporal limit of the ACOS v7.3 product as also explained in Lines 210–212 of the revised manuscript.**

**We added the explanation of the grid used as follows:**

525   Page 8, lines 200–202: *To calculate the XCO₂ that the satellite would have seen given the CO₂ profile constructed from in situ data,* ***we first interpolate these profiles onto the corresponding monthly averaged pressure grid of the ACOS and NIES retrievals, then*** *we use Eq. (15) of Connor et al. (2008):*

Page 9, lines 210–212: *Because the ACOS retrieval provides a higher number of valid data, we used the* ***pressure levels and***

530   *parameters from ACOS as representative for the calculation. After May 2016, we use the* ***pressure grid and*** *parameters from NIES due to the temporal limit of the ACOS v7.3 product.*

**Comment**

10. Page 8, line 215: You say that the values you get are similar to Matsueda et al. 2008. Can you be more specific on how

535   similar they are? You do this afterwards with the Conway et al. 1994 numbers so it would paint a more complete picture if you could also give numbers in this instance.

**Response**

**At 20° N –30° N, we found a peak-to-trough amplitudes of 6.5 ± 0.6 ppm in the upper troposphere which decreased to about 4 ppm at the equator (Lines 230–232 of the revised manuscript). Matsueda et al. (2008) found a decrease from 6**

540   **ppm at 30° N to 3 ppm at the equator. Our sentence in the manuscript was misleading as the numbers in the brackets correspond to the observations made by Matsueda et al. (2008).**

**We revised the sentence as follows:**

545   Pages 9–10, lines 234–239: *Seasonal cycles and decreasing amplitudes* ***in the upper troposphere*** *from North to South (7 ppm to 4 ppm) are similar to that observed by Matsueda et al. (2008).* ***They found a decrease from 6 ppm at 30° N to 3 ppm at the***

*equator over the same region between 2005 to 2007 using aircraft based flask samples. At sea level, seasonal cycle amplitudes that decrease from about 8 ppm at 20° N–30° N to 3 ppm at the equator were reported by the global sampling network of the National Oceanic and Atmospheric Administration's Climate Monitoring and Diagnostics Laboratory (NOAA/CMDL) (Conway et al., 1994).*

**Comment**

11. I disagree with your assessment that the mid troposphere is the only uncertainty in the in-situ profile which you need to consider. You also should consider the tropopause height as well. We can see from Figure A3 that the interpolated profile deviates from the model profile at the tropopause in panels a and c. I think your concentrations at the tropopause can't be overlooked as an uncertainty because the ACTM and your in-situ profile are clearly in disagreement over the location of the tropopause in some of these figures. It may be that the satellite data and your in-situ XCO2 are different because your tropopause height is incorrect, which would lead to seasonally dependent biases. How much does your tropopause height vary over all of the measurement times and locations in a month? By how much will this change your XCO2 values and can this account for the differences you see to the satellite data?

**Response**

**We thank Referee #2 for pointing out the uncertainty due to the tropopause height. It is correct that the variation of the tropopause height impacts the obs. $XCO_2$ and should be considered. Our results show that the data of the monthly averaged TROPPB over the ocean (see our reply to comment 4, Lines 415–438 of the response file) have the highest variation at northern midlatitudes (20° N–30° N). The standard deviation ranged from 0 to 24 hPa (0.02 to 23.77 hPa) with an average of 10 ± 5 hPa. At 0° N–10° N and 20° S–10° S, the standard deviations are lower, which ranged from 6 to 11 hPa (average 8 ± 1 hPa), and from 4 to 12 hPa (average 7 ± 2 hPa), respectively (Table R1).**

**Table R1. Range of the standard deviation (Std.) of the monthly averaged TROPPB (left column), and average standard deviation of all month (right column) in hectopascal (hPa) at each 10° latitude by 20° longitude box.**

|  | Std. range (hPa) | Average std. (hPa) |
| --- | --- | --- |
| **20° N–30° N** | 0 – 24 | 10 ± 5 |
| **0° N–10° N** | 6 – 11 | 8 ± 1 |
| **20° S–10° S** | 4 – 12 | 7 ± 2 |

**To test the effect of the TROPPB on the calculated $XCO_2$, we assumed the highest average variation of 15 hPa found at 20° N–30° N over the ocean (average standard deviation of 10 ± 5 hPa).**

**Based on observation-based $CO_2$ profiles with a TROPPB ±15 hPa, we calculated the corresponding $XCO_2$ and compared the results with our previous calculated obs. $XCO_2$. The total average difference in the obs. $XCO_2$ is only**

0.03 ± 0.06 ppm for the time period 2014 to 2017 at 20° N–30° N. Considering both the uncertainty of the TROPPB and that uncertainty caused by the variability of 2 ppm at 850 hPa in the observation-based profile (0.62 ± 0.01 ppm on average), the largest reasonable uncertainty is 0.72 ppm.

In comparison, the observed average difference between the obs. XCO$_2$ and satellite observations of ACOS v7.3 and OCO-2 v9 is 1.2 ± 0.4 ppm between June and September in 2014 to 2017. That means, even if we consider the maximal possible uncertainty of the obs. XCO$_2$ with 0.7 ppm, the discrepancy to ACOS v7.3 and OCO-2 v9 cannot be fully explained. On the other hand, the comparison of the obs. XCO$_2$ with the newer satellite retrievals ACOS v9 and OCO-2 v10 shows a reduced difference. This suggests that some bias has its origin in the older retrieval algorithm which is reduced in the newer versions (Figure A1 and A2 of the revised manuscript).

Furthermore, the sensitivity analysis, described in Lines 218–220 of the revised manuscript, showed that the impact of the upper troposphere, including the TROPPB, and stratosphere on the XCO$_2$ is only 0.2 ± 0.1 ppm on average.

**To clarify the uncertainty of the obs. XCO$_2$, we revised the text as follows:**

Page 16, lines 350–361: *It reveals that generally, the largest differences in the NH coincide with the latitude of the monthly XCO$_2$ maxima. Namely, at 30° N–40° N in spring and autumn with up to 3 ppm (between **obs.** XCO$_2$ and ACOS in March) (Figs. 6a and 6d) and in June at 10° N–20° N with a discrepancy of up to 2 ppm (between **obs.** XCO$_2$ and OCO-2) (Figs. 6b and 6e). **The difference might be caused by uncertainties in the obs. XCO$_2$ due to the variability of the TROPPB (section 3.2). However, the uncertainty in the TROPPB results in a difference in the obs. XCO$_2$ of only 0.03 ± 0.06 ppm on average. This leads to a total estimated error of 0.7 ppm considering the uncertainty of 0.62 ± 0.01 ppm derived from the ±2 ppm variation in the observation-based CO$_2$ profile at 2 km above sea level (section 3.2). It is known that a**tmospheric CO$_2$ mixing ratios in midlatitudes are characterized by high spatiotemporal variability. Therefore, the observed discrepancies in the NH may arise from differences in sample numbers, location and time within each month and latitude-longitude range. In particular, the largest uncertainty in the obs. XCO$_2$ likely results from the constructed CO$_2$ profile in the mid-troposphere, as no observational constraints are available for that part of the atmosphere and simply a linear interpolation between the ship and aircraft data was assumed (section 3.2).*

Page 16, lines 364–367: *Niwa et al. (2011) found similar straight vertical profiles between June and September in East Asia, based on aircraft observations and model results. **Furthermore, the maximum bias due to errors in the MIROC4-ACTM stratospheric CO$_2$ profile (0.9 ppm) is smaller than the average difference of 1.2 ± 0.4 ppm between the obs. XCO$_2$ and satellite observations of ACOS and OCO-2 between June and September (section 3.2).***

Page 7, lines 181–188: **The TROPPB** *is defined as a combination of a thermal tropopause- and dynamic tropopause pressure (Wilcox et al., 2012). The TROPPB data are extracted from GEOS-FP (Goddard Earth Observing System – forward*

*processing) meteorology data using the python suite "ginput" **version 1.0.6** (Laughner et al., 2021). Ginput was developed to generate a priori vertical mixing ratios of chemical species (e.g., $CO_2$, CO, $CH_4$, $N_2O$) for the open source TCCON retrieval algorithm, GGG2020 (Laughner et al., in prep).* **The TROPPB was calculated every 3 hours on the 5th, 15th, and 25th of each month for each centre location of the 10° latitude by 20° longitude boxes. Between 2014 and 2017, the highest monthly variation was found at 20° N–30° N with a standard deviation ranging from 0 to 24 hPa (0.02 to 23.77 hPa) and an average standard deviation of 10 ± 5 hPa. The maximum difference of 24 hPa at the level of the TROPPB corresponds to difference in the altitude of 1 to 2 km.**

**Comment**

12. I would be interested to see how the ACTM XCO2 would look if you interpolated it onto the satellite grid in the same way you have done for your in-situ data, and how this would compare with both the satellite and in-situ XCO2.

**Response**

**Referring to Figure A3 (Figure A4 of the revised manuscript) as mentioned in comment 11, we interpolated the monthly averaged ACTM $CO_2$ profiles onto the 20 pressure levels of GOSAT ACOS for June and July 2014 and 2015, and for June and July 2016 onto the 15 pressure levels of GOSAT NIES. At 20° N–30° N, the variation of the monthly averaged ACTM $CO_2$ data varies from 0.6 % (2.4 ppm) at surface level to less than 0.02% (0.07 ppm) at the upper troposphere from 2014 to 2017. We assumed an average variation of 0.3% at each of the 67 pressure layers of the ACTM. The comparison of the ACTM $CO_2$ profile with that interpolated on the ACOS grid shows mainly differences in the upper troposphere which is explained by the lower resolution of the ACOS grid (Fig. R6 and Fig. R7).**

**The resulting $XCO_2$ is within 1 ppm (average difference 0.20 ± 0.32 ppm) of that of the obs. $XCO_2$. The highest average difference to the satellite retrievals occur for ACOS and OCO-2 with 1.11 ± 0.28 ppm and 1.02 ± 0.40 ppm, respectively (Table R2 and R3). The results reveal that $XCO_2$ based on the ACTM show a similar difference to $XCO_2$ from ACOS and OCO-2 as the obs. $XCO_2$ (compare Table 3 in the revised manuscript).**

[Figure]

Figure. R6. Comparison of the $CO_2$ profiles of the ACTM (green) with that interpolated onto the 20 pressure levels of ACOS (black). Error bars are the standard deviation of the monthly averages of the ACTM. Grey lines represent 0.3% uncertainty of the interpolated ACTM.

[Figure]

Figure. R7. Similar to Fig. R6, but with the y-limits set to focus on the upper troposphere. Comparison of the $CO_2$ profiles of the ACTM (green) with that interpolated onto the 20 pressure levels of ACOS (black). Error bars are the standard deviation of the monthly averages of the ACTM. Grey lines represent 0.3% uncertainty of the interpolated ACTM.

**Table R2. The monthly averaged (Avg.) values and its standard deviation of ACTM based XCO₂, obs. XCO₂ and satellite XCO₂ of GOSAT (ACOS, NIES) and OCO-2 in the latitude range 20° N–30° N, for June and July, 2014–2017.**

| Date (MM/YY) | Avg. ACTM XCO₂ (ppm) | Avg. obs. XCO₂ (ppm) | Avg. ACOS (ppm) | Avg. NIES (ppm) | Avg. OCO-2 (ppm) |
|---|---|---|---|---|---|
| 6/14 | 400.65 ± 1.07 | 400.7 ± 0.62 | 399.78 ± 1.24 | 401.49 ± 1.64 | – |
| 7/14 | 398.79 ± 1.06 | 398.8 ± 0.63 | 397.93 ± 1.12 | 399.57 ± 0.7 | – |
| 6/15 | 402.47 ± 1.08 | 402.87 ± 0.63 | 401.34 ± 1.20 | 402.67 ± 0.71 | 401.35 ± 0.69 |
| 7/15 | 401.01 ± 1.06 | 400.82 ± 0.62 | 399.45 ± 1.20 | 401.06 ± 1.39 | 399.78 ± 0.97 |
| 6/16 | 405.48 ± 1.10 | 405.62 ± 0.63 | – | 405.49 ± 0.96 | 404.1 ± 0.69 |
| 7/16 | 403.26 ± 1.12 | 404.06 ± 0.63 | – | 401.85 ± 3.74 | 402.91 ± 1.28 |

645 **Table R3. Difference between the average values of ACTM based XCO₂ and the obs. XCO₂ and satellite XCO₂.**

| Date (MM/YY) | ACTM − in situ XCO₂ (ppm) | ACTM − ACOS XCO₂ (ppm) | ACTM – OCO2 XCO₂ (ppm) | ACTM − NIES XCO₂ (ppm) |
|---|---|---|---|---|
| 6/14 | −0.05 | 0.87 | – | −0.84 |
| 7/14 | −0.01 | 0.86 | – | −0.78 |
| 6/15 | −0.4 | 1.10 | 1.1 | −0.2 |
| 7/15 | 0.2 | 1.60 | 1.2 | −0.1 |
| 6/16 | −0.14 | – | 1.38 | −0.01 |
| 7/16 | −0.8 | – | 0.35 | 1.41 |

**Comment**

13. Page 15, line 354: It is unclear to me how satellite XCO2 can show a delayed response to CO2 changes since it is measuring in real time. Please could you elaborate on this.

650 **Response**

**We agree, the argument needs to be clarified. Referee #1 criticized the same point. What we observed is that the satellite retrievals sometimes show the extreme values of XCO₂ delayed by one month as compared to our obs. XCO₂ dataset. We know from previous long-term in situ measurements in the upper troposphere and at surface level (Lines 382–385 of the revised manuscript) that maxima and minima of CO₂ occur not later than May and September, respectively.**

655 **Therefore, maxima in June and minima in October of the satellite retrievals are later than observed by the in situ measurements.**

**Satellite XCO₂ is measured in real time. We argue that there are remaining uncertainties which are introduced by limitations in the retrieval algorithm, which have not been previously identified due to the lack of validation data over the open ocean. We know that from GOSAT and OCO-2 the retrieval algorithm to obtain XCO₂ from the measured**

660 **radiance are undergoing rapid progress with almost one new version per year for OCO-2. We are hoping that the highly accurate ship and aircraft data over a unique geographical region will help us to build the capacity for the validation and improvement of satellite XCO₂ retrievals.**

**We clarified our statement as follows:**

665

Page 17, lines 385–388: *The consistency with long-term studies support the correctness of the* **obs.** *XCO₂, which implies that satellite XCO₂ sometimes show a delayed response to CO₂ changes**, which might be caused by remaining uncertainties introduced by limitations in the retrieval algorithms and have not been previously identified due to the lack of validation data over the open ocean.*

Page 19, lines 441–444: *Hence, the result indicates that even if the retrievals complement each other, measurement uncertainties remain, which limit the accurate interpretation of spatiotemporal changes in $CO_2$ fluxes by satellites alone.* ***These uncertainties might be introduced by limitations in the retrieval algorithms and have not been previously identified due to the lack of validation data over the open ocean.***

**Comment**

14. Page 16, line 359: Please add the uncertainty and be more accurate in your "less than 2ppm" number since you are being very precise with the number you are comparing it to in the next sentence.

**Response**

 **We revised the sentence as follows:**

Page 18, lines 391–392: *From 2014 to 2015,* ***obs.*** *and satellite XCO₂ increased by* ***1.61 ± 0.24 ppm yr⁻¹ on average*** *at 20° N–30° N (**Table 4,** Fig. 5a).*

 **Comment**

15. Page 16, line 360: 3.84  0.65 and 3.39  0.03 overlap so you can't draw conclusions about their difference.

**Response**

**Referee #2 is correct that the uncertainties overlap. This was also pointed out by Referee #1. The difference in the increase of the obs. XCO₂ and that of the satellite XCO₂ isn't significant at northern latitudes, but the increase of the**  **obs. XCO₂ tends to be slightly higher. At the equator, the increase of the obs. XCO₂ is significantly higher than that of ACOS and OCO-2 (two-sided t-test, significance level α=0.05). We revised this part as follows:**

Page 18, lines 392–396: *In contrast, a significant increase of 3.84 ± 0.65 ppm yr⁻¹ is observed by* ***obs.*** *XCO₂ from 2015 to 2016.* ***The average increase of the mean values of all satellite retrievals is 3.39 ± 0.03 ppm yr⁻¹.*** *This rapid increase is also*  *seen near the equator,* ***where the increase of the obs. XCO₂ is significantly higher than that of ACOS and OCO-2 (two-sided t-test, significance level α=0.05). S****imultaneously*, *a larger negative bias of the satellite XCO₂ in 2016 as compared to the previous years* ***is observed*** *(Figs. 5b and 5e).*

**Comment**

700 16. You conclude by saying that datasets such as yours can augment TCCON data. Did you consider applying your method to other aircraft and ground data at TCCON sites to compare with TCCON itself?

**Response**

**In future, we plan to extend the temporal coverage, and increase the temporal and spatial resolution. Then we can also compare our method to other aircraft and also to ground data from TCCON stations to strengthen our results.**

705 **Currently, we couldn't compare our data with other aircraft campaign such as $CO_2$ profiles from HIPPO (Hiaper Pole-to-Pole Observations) or ATom (Atmospheric Tomography Mission). This is because no coincident data between the years 2014 and 2017 in the longitude–latitude range of 130° E to 173° E and 30° S to 40°N exist. It would be beneficial to compare our results with data of coastal or island TCCON stations to test our methodology. But again, a direct comparison is currently not possible as no co-located CONTRAIL (aircraft) and ship measurements with TCCON**

710 **stations are available. For example, for the latitude range of the TCCON station Wollongong, we don't have aircraft data from over the ocean (compare our response to Referee#1, Lines 31–44 of the response file).**

**Our immediate plan is to compare our results with ship based EM27/Sun Fourier Transform infrared spectrometer measurements, which were conducted this spring on the research vessel Mirai in collaboration with the research group of André Butz of the University of Heidelberg.**

715

**Technical corrections**

**Comment**

Page 2, line 45: You say that most sites are in North America and Europe, and some are in East Asia and Oceania. This reads

720 as though there are no sites in the other continents, which I don't think was your intention to say. Please reword this sentence.

**Response**

**We revised the sentence as follows:**

Page 2, lines 46–48: *There are now more than 100 surface measurement sites around the globe, but most are located on land*

725 *in North America and Europe, and some in the East Asia and Oceania, **and few in other continents** (e.g., Crowell et al., 2019; Hakkarainen et al., 2019).*

**Comment**

Page 2, line 55: "improves" should be "improve".

**Response**

**The sentence is corrected as follows:**

Page 2, lines 55–57: *These observations are most sensitive to the lower troposphere where $CO_2$ is most variable (Patra et al., 2003) and therefore, are able to improve the knowledge on local $CO_2$ emission and sinks (Connor et al., 2008).*

**Comment**

Page 3, line 74: "Aircrafts" should be "aircraft".

**Response**

**The sentence is corrected as follows:**

Page 3, lines 76–78: *In combination with surface measurements, vertical profiles of $CO_2$ obtained by aircraft can constrain $XCO_2$ but are very limited (e.g., Frankenberg et al., 2016; Inoue et al., 2013; Wofsy, 2011; Wofsy et al., 2018).*

**Comment**

Page 3, line 79: Negative symbol isn't the same type of symbol used in previous cases.

**Response**

**The sentence is corrected as follows:**

Page 3, lines 81–83: *More recent comparisons of OCO-2 $XCO_2$ estimates to in situ measurements from the NASA Atmospheric Tomography Mission reveals a systematic bias of $-0.7$ ppm over the tropical Pacific, that is also seen in **the** data **at Burgos, a TCCON station** in that region **(Kulawik et al., 2019; Velazco et al., 2017).***

**Comment**

Page 3, line 80: Remove journal and doi from citation.

**Response**

**The sentence is corrected as follows:**

Page 3, lines 81–83: *More recent comparisons of OCO-2 XCO$_2$ estimates to in situ measurements from the NASA Atmospheric Tomography Mission reveals a systematic bias of −0.7 ppm over the tropical Pacific, that is also seen in **the** data **at Burgos,**
**a TCCON station** in that region **(Kulawik et al., 2019; Velazco et al., 2017).***

Page 28, lines 598–602: **Kulawik, S. S., Crowell, S., Baker, D., Liu, J., Mckain, K., Sweeney, C., Biraud, S. C., Wofsy, S., Dell, C. W. O., Wennberg, P. O., Wunch, D., Roehl, M., Deutscher, N. M., Kiel, M., Griffith, D. W. T., Velazco, V. A., Dubey, M. K., Sepulveda, E., Elena, O., Rodriguez, G., Té, Y., Heikkinen, P., Dlugokencky, E. J., Gunson, M. R., Eldering, A., Fisher, B. and Osterman, G. B.: Characterization of OCO-2 and ACOS-GOSAT biases and errors for CO$_2$ flux estimates, (October), 2019.**

Page 30, lines 678–680: **Velazco, V. A., Morino, I., Uchino, O., Hori, A., Kiel, M., Bukosa, B., Deutscher, N. M., Sakai, T., Nagai, T., Bagtasa, G., Izumi, T., Yoshida, Y. and Griffith, D. W. T.: TCCON Philippines: First measurement results, satellite data and model comparisons in Southeast Asia, Remote Sens., 9(12), 1–18, doi:10.3390/rs9121228, 2017.**

**Comment**

Page 8, line 210: Missing "N" in "20 –30 N" whilst in other instances you have used "20 N–30 N".

**Response**

**The sentence is corrected as follows:**

Page 9, lines 230–231: *At 20° **N**–30° N, the peak-to-trough amplitudes of the seasonal cycles at sea level is 8.5 ± 0.9 ppm, and is ~2 ppm larger than the amplitudes in the upper troposphere (6.5 ± 0.6 ppm).*

**Comment**

Figure A2: Add an x-label.

**Response**

**The Figure A2 corresponds to Figure A3 of the revised manuscript. The x-label is revised as follows:**

[Figure]

785    Page 23, lines 467–468: Figure A**3**. Latitude-pressure distribution of the inversion of the $CO_2$ mixing ratio at longitude 146° E in 2015, obtained from ACTM forward simulations.

**Other corrections**

**We corrected a few values caused by a mistake in the calculation script, some typos, and inconsistent spellings as**
790  **follows:**

Page 2, lines 49–51: *The uneven distribution and limited spatial coverage of in situ measurements makes it **difficult** to infer $CO_2$ fluxes between the surface and the atmosphere on regional to global scales **at high accuracy** (Canadell et al., 2011; Chevallier et al., 2010, 2011).*

795

Pages 5–6, lines 151–160: *The details of the MIROC4-ACTM are described in Patra et al. (2018). In short, the MIROC4-ACTM uses a hybrid vertical coordinate to resolve gravity wave propagation in the stratosphere**, where at least 30 model layers reside**. The hybrid coordinate transitions from sigma coordinates at the surface to pressure levels around the tropopause. **In total, 67 vertical layers are used between the Earth's surface and 0.0128 hPa. The MIROC4-ACTM has a**
800  **horizontal resolution of triangular 42 truncation (T42) which corresponds to approximately 2.8° longitude by 2.8° latitude.** The ACTMs are nudged with the Japanese 55-year Reanalysis (JRA-55; Kobayashi et al., 2015) for horizontal winds and temperature at Newtonian relaxation times of 1-hour and 5-hours, respectively. Nudging is performed for all the model layers from 2 to 60. A high accuracy of the MIROC4-ACTM is indicated by the agreement of simulated "age of air", which is a diagnostic for atmospheric transport, with that expected from measured sulphur hexafluoride ($SF_6$) and $CO_2$ in the troposphere*
805  *and stratosphere, respectively (Patra et al., 2018). All data obtained over land are excluded **in the current study**.*

Page 9, lines 219–220: *The difference in $XCO_2$ was **only** 0.2 ± 0.1 ppm on average.*

Page 14, lines 324–325: *In all latitudes, **obs.** and satellite $XCO_2$ show an overall significant positive correlation ($R^2$: NIES =*
810  *0.84 ± 0.02, ACOS = 0.74 ± 0.0**8**, OCO-2 = 0.8**2** ± 0.0**5**) (Table 2).*

Page 14, lines 327–329: *The smallest **average** bias is found for NIES, likely due to the stricter quality filters as discussed in section 4.1. **While ACOS and OCO-2 show rather a systematic offset, the NIES retrieval seems to be more noisy (Figs. 5d and 5e, Table 3).***
815

Page 14, lines 329–330*: ... and decreases by 40% (0.5**6** ppm) on average between the northernmost and southernmost regions (Table 2).*

820    Page 15, line 344:

| | $R^2$ | | | RMSE | | |
|---|---|---|---|---|---|---|
| Latitude | NIES | ACOS | OCO-2 | NIES | ACOS | OCO-2 |
| 20° N–30° N | 0.86 | 0.64 | 0.81 | 1.06 | 1.70 | 1.26 |
| 0° N–10° N | 0.81 | 0.76 | 0.76 | 1.02 | 1.17 | 1.23 |
| 20° S–10° S | 0.8**5** | 0.8**2** | 0.8**8** | 0.84 | 0.**79** | 0.7**0** |

Page 16, lines 346–348: *Table 3. Average (Avg.) difference and the standard deviation (S**D**.) between **obs.** and satellite $XCO_2$ from GOSAT (NIES, ACOS) and OCO-2 of each latitude range between 2014 and 2017.*

| | difference **obs**. $XCO_2$ – satellite $XCO_2$ | | | | | |
|---|---|---|---|---|---|---|
| Latitude | Avg. NIES | S**D**. | Avg. ACOS | S**D**. | Avg. OCO-2 | S**D**. |
| 20° N–30° N | 0.61 | 0.87 | 1.60 | 0.59 | 1.14 | 0.52 |
| 0° N–10° N | 0.51 | 0.87 | 1.00 | 0.60 | 1.12 | 0.52 |
| 20° **S**–10° S | 0.2**0** | 0.8**1** | 0.48 | 0.63 | 0.31 | 0.63 |

825    Page 2, lines 63–64: *Since 2009, NASA**'s** Atmospheric $CO_2$ Observation from Space (ACOS) and GOSAT team work closely together on the analysis of GOSAT observations (Crisp et al., 2012; O'Dell et al., 2012).*

Page 3, lines 79–81: *Comparisons of ACOS GOSAT $XCO_2$ estimates to those from HIAPER Pole-to-Pole Observations (HIPPO) campaigns (Frankenberg et al., 2016) show lower bias (−0.06 ppm) and standard deviation (0.45 ppm).*

830

Page 5, lines 136–137: *For ACOS and OCO-2, we chose data with a good quality flag (quality flag = 0), which is provided by each algorithm.*

Page 9, lines 226–228: *Average $CO_2$ mixing ratios of 402.9 ± 3.6 ppm and 401.2 ± 3.1 ppm at lower and upper troposphere,*

835    *exceeded that from south of the equator by 4.5 ppm and 1.5 ppm, respectively.*

Pages 11–12, lines 277–278: *The coarse horizontal resolution of the model is not adequate to represent observations near source regions.*

840    Lines 196, 274, 279, 363: Changed from *MIROC-4 ACTM* to ***MIROC4-ACTM***

Lines 163, 224, 243: Changed *mid latitudes* to ***midlatitudes***

Lines 175, 228: 236, 273, 277, 280, 300: Changed *sea-level* to **sea level**

845

Lines 295, 296, 297, 323, 327, 331, 413: Changed *Fig.* to **Figs.**

[revised manuscript text omitted]